# Azoles activate type I and type II programmed cell death pathways in crop pathogenic fungi

Martin Schuster [1,2], Sreedhar Kilaru [1,2] & Gero Steinberg [1] ✉

Triazoles are widely used to control pathogenic fungi. They inhibit the ergosterol biosynthetic pathway, but the precise mechanisms leading to fungicidal activities in many fungal pathogens are poorly understood. Here, we elucidate the mode of action of epoxiconazole and metconazole in the wheat pathogen *Zymoseptoria tritici* and the rice blast fungus *Magnaporthe oryzae*. We show that both azoles have fungicidal activity and reduce fluidity, but not integrity, of the plasma membrane. This impairs localisation of Cdc15-like F-BAR proteins, resulting in defective actin ring assembly and incomplete septation. However, mutant studies and pharmacological experiments in vitro and *in planta* show that azole lethality is due to a combination of reactive oxygen species-induced apoptosis and macroautophagy. Simultaneous inhibition of both programmed cell death pathways abolishes azole-induced cell death. Other classes of ergosterol biosynthesis inhibitors also induce apoptosis and macroautophagy, suggesting that activation of these two cell death pathways is a hallmark of ergosterol synthesis-targeting fungicides. This knowledge will inform future crop protection strategies.

Fungi pose serious threats to the world's food security[1]. It was estimated that severe epidemics in the five most important crops, namely wheat, rice, maize, potato and soybean, could destroy ~60% of the world's food supply[2]. Fungal diseases are mitigated by management strategies, which include biological and RNAi-based control and genetically modified disease resistant cultivars[3]. However, our most potent 'weapon' against fungal crop pathogens are fungicides. Indeed, the increasing demand for food made modern agriculture reliant on these anti-fungal chemicals[4]. The most important class of agricultural fungicides are triazoles (hereafter named azoles), which combine low cost, reliable performance and broad effectiveness[4]. Consequently, azoles hold 20–25% of the world agricultural fungicide market[4,5], worth USD 3.3–4 billion (www.grandviewresearch.com/industry-analysis/fungicides-market).

Azoles interfere with the modification of sterols during the ergosterol biosynthetic pathway (FRAC code 3; https://www.frac.info). They directly bind and inhibit 14α-lanosterol demethylase, which is a key enzyme in the ergosterol biosynthetic pathway. Inhibition of 14α-lanosterol demethylase results in ergosterol depletion in fungal membranes, which is expected to change lipid organisation and membrane fluidity[6]. In addition, toxic intermediates are formed[7], which further the impact on membrane structure and function[8]. These effects are generally thought to inhibit growth (=fungistatic activity[9]), rather than killing the pathogen[8,10]. However, delayed killing of the fungal cell (=fungicidal activity[9]) was also reported for some fungi[11,12].

Ergosterol is enriched in the fungal PM[13], and it is assumed that azoles compromise the integrity of this outer cell boundary[14,15]. Indeed, treatment of *Candida albicans* with itraconazole resulted in potassium ion leakage from the cell, although no major disruption of the PM was found[10]. However, ergosterol is also present in various organelles[13], and azole toxicity was related to impaired organelle function, including disruption of vacuolar ATPase activity[16], mitochondrial dysfunction, accompanied with toxic reactive oxygen species (ROS) development[17,18] and apoptotic programmed cell death[10], and late

[1]Biosciences, University of Exeter, EX4 4QD Exeter, UK. [2]These authors contributed equally: Martin Schuster, Sreedhar Kilaru.
✉e-mail: G.Steinberg@exeter.ac.uk

endosomal trafficking[19]. In addition, azoles are thought to exert their antifungal activity via altered cell wall synthesis and organisation[11,20], changes in the actin cytoskeleton[21] and DNA damage[22]. The broad range of physiological impact of azoles mask their effect on cells, underpinning the anti-fungal activity of azoles.

Our understanding of the impact of azoles on fungal cells is almost entirely restricted to human pathogens, including *Candida* species, *Aspergillus fumigatus*, *Cryptococcus neoformans* and various dermatophytes. How azoles affect plant pathogenic fungi remains elusive. Here we investigate the effect of epoxiconazole and metconazole, azoles that are widely used in crop protection[4,5], in the wheat pathogen *Zymoseptoria tritici* and the rice blast fungus *Magnaporthe oryzae*, both considered to be important crop pathogens worldwide[23]. We show that azoles and other ergosterol biosynthesis inhibitors have a common impact (mode of action, MoA) in these crop pathogens; they induce apoptosis and macroautophagy (hereafter named autophagy). We further show that combined activation of both cell death pathways underpins the fungicidal activity of azoles in plant pathogens.

## Results

### Azoles are initially fungistatic, but become fungicidal with time

In this study, we used a cell biological approach to investigate the physiological effects of the azoles epoxiconazole and metconazole, used as unformulated and pure compounds, in the wheat pathogen *Z. tritici* (Fig. 1a, Supplementary Fig. 1a, plasma membrane [=PM] is labelled with a fusion protein of red-fluorescent mCherry and the *Z. tritici* a syntaxin-like protein [name: mCherry-Sso1]; nuclei labelled with the *Z. tritici* homologue of the DNA-binding histone1, fused to codon-optimised green-fluorescent protein[24] [name: His1-ZtGFP]; for genotype of strains and plasmids see Supplementary Tables 1 and 2; for experimental usage of all strains see Supplementary Table 3; colour-blind friendly images are provided in the Supplementary Information). We began our study by asking which effect epoxiconazole and metconazole has in *Z. tritici*. In a first set of experiments, we investigated the effect of azoles on colony formation on agar plates. We found that both azoles reduced growth by half at 0.003–0.007 µg ml⁻¹ and prevented any visible growth at -0.007–0.011 µg ml⁻¹ (=minimal inhibitory concentration, MIC; Supplementary Fig. 2a, b).

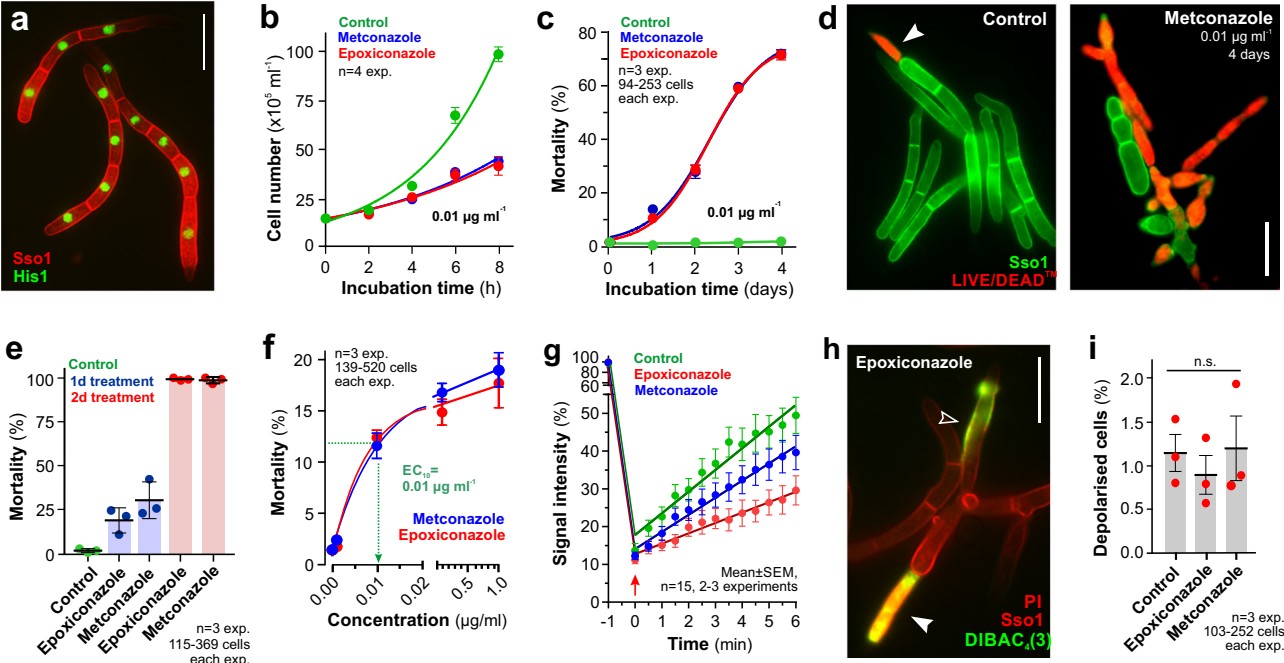

**Fig. 1 | Effects of azoles on the plasma membrane in *Z. tritici*. a** Multi-cellular conidia of *Z. tritici*, expressing the plasma membrane marker mCherry-ZtSso1 (red, Sso1) and the nuclear marker His1-ZtGFP[24] (green, His1). Scale bar = 10 µm. **b** Growth of *Z. tritici* cells in the presence of the solvent (green), 0.01 µg ml⁻¹ epoxiconazole (red) and 0.01 µg ml⁻¹ metconazole (blue). Note that cells are inhibited in growth, suggesting a fungistatic activity of both azoles. Sample size $n = 4$. See Supplementary Fig. 3a for effect of higher concentrations. **c** Mortality of *Z. tritici* in the presence of the solvent (Control) and 0.01 µg ml⁻¹ azoles. Mortality was determined using LIVE/DEAD™ staining. Note that -70% of all cells are dead after 4 days of incubation with low concentrations of azoles. Sample size $n = 3$. See Supplementary Fig. 3b, c for effect of higher concentrations. **d** LIVE/DEAD™ staining (red) of *Z. tritici* conidia, expressing the plasma membrane marker eGFP-ZtSso1 (green, Sso1), after 4 days incubation with 0.01 µg ml⁻¹ epoxiconazole. **e** Mortality after treatment with epoxiconazole or metconazole for 1 day (blue) or 2 days (red), followed by growth for 3 (blue) or 2 days (red) in azole-free medium. Control cells were treated with the solvent methanol. Arrowhead points towards a dead cell. Sample size $n = 3$. **f** Relative mortality of *Z. tritici* cells after 24 h treatment with various concentrations of epoxiconazole and metconazole. Mortality was assessed using LIVE/DEAD™ staining. The effective concentration at 10% increase of mortality over background mortality in control sample (effective concentration at 10%; EC₁₀) is indicated in green. Sample size $n = 3$. **g** Fluorescent recovery after photo-bleaching eGFP-Sso1 in *Z. tritici* cells, treated with the solvent (Control) and azoles. All linear regression curves fit at $R^2 > 0.956$; both azole recovery slopes are significantly different from control at $P < 0.0001$. Sample size $n = 15$ cells from 2-3 experiments. See Supplementary Movie 1. **h** Ion permeability of the plasma membrane in epoxiconazole-treated cells. Depolarised cells accumulate DiBAC₄(3) (green; open arrowhead); propidium iodide-positive cells (red, PI) indicated by filled arrowhead; plasma membrane labelled with mCherry-ZtSso1 (red, Sso1) Scale bar = 10 µm. **i** Relative number of depolarised cells after 24 h treatment with 0.01 µg ml⁻¹ epoxiconazole, metconazole or the solvent methanol (Control). Propidium iodide-positive cells were excluded. Sample size $n = 3$. Cells were grown in YG media at 18 °C, 200 rpm, and treated with various concentrations (**b**), or with 0.01 µg ml⁻¹ (**c, d, e, g, h, i**) and 1 µg ml⁻¹ (**f**) and grown for 2 h to 4 days; in all control experiments, 0.1% methanol was used (v v⁻¹). Results shown in (**a, d, h**) were obtained independently in three experiments. Values in (**b, c, d, e, f, g, i**) are mean ± standard error of the mean (SEM); dots in (**e, i**) represent average of independent experiments; statistical testing in (**i**) used one-way ANOVA testing; n.s.: not significantly different at two-tailed $P > 0.05$; linear and non-linear regression and slope comparison in (**b, c, d, g**) was done in Prism 6. Colour-blind friendly images are available in the Supplementary Fig. 1. All data are provided in the Source Data File.

Azoles are generally considered to inhibit fungal growth without killing the cell, although some fungicidal activity was reported[25]. To test for such activity in *Z. tritici*, we incubated cells in liquid culture with 0.01 µg ml⁻¹ for various time points and (i) investigated cell proliferation by counting individual cells in the multi-cellular spores, which were defined by the fluorescent PM marker Sso1, and (ii) determined cell killing by quantitative LIVE/DEAD™ staining (for technical details see Methods). This membrane integrity-based cell viability assay is based on an amine-reactive dye that binds covalently to proteins inside cells with compromised membranes, thereby yielding intense red-fluorescent staining in dead *Z. tritici* cells[26]. We found that incubation with 0.01 µg ml⁻¹ azoles cause growth inhibition during the first 8 h incubation (Fig. 1b, Supplementary Fig. 1b). However, with time of azole exposure an increasing number of red-fluorescent dead cells was found and mortality reached ~70% after 4 days (Fig. 1c, d, Supplementary Fig. 1c, d; red/magenta colour in 1d indicates cell death). Thus, in *Z. tritici* azoles tested were initially fungistatic and became fungicidal after days of incubation. This confirms previous reports in *A. fumigatus*[11]. We next tested if this effect is due to azole concentration. We applied 100 times higher azole concentrations (1 µg ml⁻¹) and again found that initial fungistatic activity is followed by killing of the pathogen cell. However, the mortality was increased and reached almost 100% after 4 days (Supplementary Fig. 3a–c). Thus, while the fungistatic activity is independent of the applied azole concentration, higher concentrations increase fungicidal activity in *Z. tritici*.

We next asked if the fungicidal activity requires extended exposure to azoles. We incubated cells with 1 µg ml⁻¹ azoles for 1 or 2 days, washed the cells with fresh azole-free medium and continued incubation in the absence of azoles. After 4 days, we determined cell mortality by LIVE/DEAD™ staining. We found that incubation for 1 day, followed by 3 days in fresh medium, resulted in ~20–30% dead cells (Fig. 1e, Supplementary Fig. 1e), corresponding to the mortality seen at 1 day (Supplementary Fig. 3b; Student's *t* test not significantly different at two-tailed *P* = 0.4855 [epoxiconazole] and *P* = 0.2727 [metconazole]). Thus, cells did neither recover, nor continue dying after removal of the fungicides. However, when azoles were removed after 2 day-exposure, mortality after 4 days was not ~65%, as measured at 2 days (Supplementary Fig. 3b), but rather increased to 100% (Fig. 1e, Supplementary Fig. 1e). This result suggests that cells reached a 'point of no return' after 2 day-exposure. This result provides first indication that azoles initiated a death programme in the pathogen.

## Azole treatment alters the fluidity but not the integrity of the plasma membrane

We investigated the mechanism by which azoles kill the pathogen cell (mode of action, MoA). Previous work has shown that the primary physiological effect of fungicides can be determined at low concentrations, where mortality is at ~10–20% and most of the cells are still alive[27]. We established that 24 h treatment with 0.01 µg ml⁻¹ azoles induce 10% mortality (Fig. 1f, Supplementary Fig. 1f; note that this concentration corresponds with the MIC, see above) and thus used this condition for all subsequent experiments in *Z. tritici*.

In the yeast *Saccharomyces cerevisiae*, ergosterol is concentrated in the PM[13], and azoles are generally thought to act on the integrity of the outer membrane[14,15]. We firstly tested if azoles influence PM fluidity by monitoring fluorescent recovery after photo-bleaching of the fluorescent syntaxin eGFP-Sso1 in *Z. tritici* cells. We found that azole treatment decreased the mobility of fluorescent eGFP-Sso1 (Fig. 1g, Supplementary Fig. 1g; Supplementary Movie 1), suggesting that inhibition of ergosterol synthesis affects the composition of the PM in *Z. tritici*. To test if azoles impair PM integrity, we stained azole-treated *Z. tritici* cells with the voltage-sensitive fluorescent probe bis-(1,3-dibutylbarbituric acid) trimethine oxonol (DiBAC₄(3)). This dye only enters depolarised cells that leaked ions through their PM (Fig. 1h,

Supplementary Fig. 1h, open arrowhead) and was used previously to investigate the integrity of the PM in *Z. tritici*[26]. As DiBAC₄(3) also stains dead *Z. tritici* cells, we identified these by co-staining with propidium iodide and excluded them from the analysis (Fig. 1h, Supplementary Fig. 1h, filled arrowhead; note that the cell viability LIVE/DEAD™ dye was interfering with most staining assays and thus was replaced by red-fluorescent propidium iodide, which binds to DNA and thus also detects PM-disrupted dead *Z. tritici* cells[26]). This revealed that 24 h treatment with 0.01 µg ml⁻¹ azoles did not significantly increase the number of DiBAC₄(3)-positive depolarised cells (Fig. 1i, Supplementary Fig. 1i). Thus, we conclude that azoles at low concentrations do not increase the ion permeability of the cell, suggesting their fungicidal activity in *Z. tritici* is not related to a disruption of the PM.

## Azole treatment results in septation defects

Treatment with azoles resulted in shorter and more compact cells within the multi-cellular conidia (Fig. 2a, Supplementary Fig. 4a, only epoxiconazole-treated cells shown). This is illustrated by a significantly reduced cell polarity index, calculated as the quotient of cell length divided by cell width (Fig. 2b, Supplementary Fig. 4b). Azole treatment also resulted in mis-placed septae and smaller cellular compartments (Fig. 2a, Supplementary Fig. 4a, inset) with reduced volume (Fig. 2c, Supplementary Fig. 4c; a cylindrical cell shape was assumed). In addition, eGFP-Sso1-labelled septa often appeared incomplete (Fig. 2d, Supplementary Fig. 4d; Supplementary Movie 2). Ultrastructural studies showed that incomplete septa lacked Woronin bodies[28] (Fig. 2e, Supplementary Fig. 4e), required to seal the septal pores upon cell injury to protect adjacent cells. To test if septae cannot be sealed, we wounded cells of a strain IPO323_His1ZtG_mChSso1, which expresses mCherry-Sso1 and the nuclear marker His1-ZtGFP[24]. In control conditions, disruption of distal cell in a conidium led to leakage of its cytoplasm and nucleus, whereas nuclei in adjacent cells remained stationary (Supplementary Movie 3; Control). However, in epoxiconazole-treated cells, wounding the tip cell resulted in collapse of subapical cells and movement of nuclei through septa, even when these appeared intact in fluorescent microscopy images (Fig. 2f, Supplementary Fig. 4f; asterisk). Based on the movement of nuclei, we estimated that ~40–60% of all septa are unable to seal after wounding (Fig. 2g, Supplementary Fig. 4g; Supplementary Movie 3). This result confirmed that azole treatment results in incomplete septation.

Fungal septum formation is driven by the inward constriction of a PM-associated F-actin ring[28]. We visualised F-actin rings in azole-treated *Z. tritici* cells, using a strain that expressed the fluorescent actin reporter Lifeact-ZtGFP[24]. In control cells, peripheral cytokinetic F-actin rings were found (Fig. 2h, Supplementary Fig. 4h, Early stage) that constricted and formed an actin-rich 'plate' at later stages of septation[28] (Fig. 2h, Supplementary Fig. 4h, Late stage). In contrast, azole-treated cells contained incomplete and often 'patchy' F-actin rings (Fig. 2i, Supplementary Fig. 4i; Supplementary Movie 5). These results suggest that changes in the ergosterol content of the PM affect F-actin-ring formation, which in turn results in septation defects. In the fission yeast, the F-actin ring is organised by Cdc15 and Imp2, which bridge between the PM and the F-actin ring via their F-BAR domain[29,30]. We identified the putative protein ZTRI_2.238 in *Z. tritici* (https://fungidb.org/fungidb/app/record/dataset/NCBITAXON_336722), which contains a F-BAR domain (InterProScan; www.ebi.ac.uk/interpro/search/sequence/) and shares 40.4% sequence identity with *S. pombe* Imp2 and 33.5% sequence identity with *S. pombe* Cdc15. We generated a strain that expressed mCherry-ZtSso1 and a fusion protein of the putative F-BAR protein (named ZtImp2) and codon-optimised green fluorescent protein[24]. Co-visualisation of the red-fluorescent PM and ZtImp2-ZtGFP revealed that the fusion protein is located at growing septa (Fig. 2j, Supplementary Fig. 4j, 5a). However, it was largely absent from incomplete septa in epoxiconazole-treated cells (Fig. 2j, Supplementary Figs. 4j, 5b; Supplementary Movie 4). This suggests that

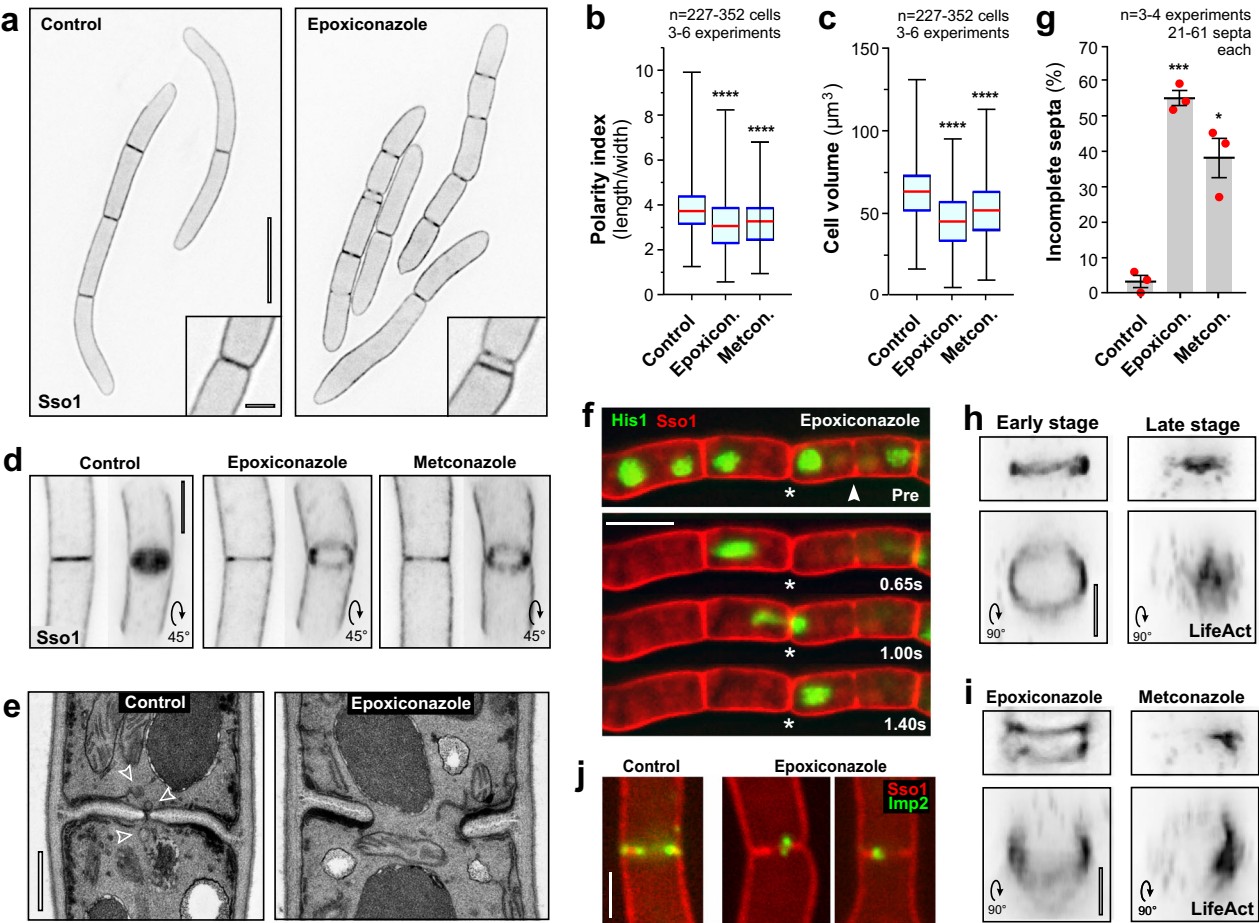

**Fig. 2 | Azoles induce incomplete septation. a** Morphology defect in epoxiconazole-treated *Z. tritici* conidia. The plasma membrane is labelled with eGFP-ZtSso1 (indicated by Sso1); images are contrast inverted. Inserts show mis-positioned septa. Scale bars = 10 μm (overview), 2 μm (inset). **b** Polarity index of control and azole-treated cells. Sample size *n* = 227–352 cells from three to six independent experiments. **c** Cell volume in cells treated with solvent (control) or azoles. Calculations assumed a cylindrical shape ($V = \pi r^2 h$). Sample size *n* = 227–352 cells from 3 to 6 independent experiments. **d** eGFP-ZtSso1-labelled septa in solvent- (Control) and azole-treated cells. Images are contrasted inverted. Tilting 3D reconstructions (45°) show open septa. Scale bar = 3 μm. See Supplementary Movies 2. **e** Ultrastructure of septa in solvent- (control) and epoxiconazole-treated cells (epoxiconazole). Woronin bodies are indicated by arrowheads. Scale bar = 0.5 μm. **f** Movement of His1-ZtGFP-labelled nuclei (green; His) through mCherry-ZtSso1-labelled septa (red; Sso1) after laser-wounding of an epoxiconazole-treated conidium. Wounding point is located to the right and not shown. Time after wounding in seconds is indicated; 'Pre': before laser treatment; asterisk shows septum that appears intact; arrowhead shows incomplete septum. Scale bar = 5 μm. See Supplementary Movie 3. **g** Number of 'open' septa identified by passage of nuclei after laser wounding in solvent (Control) and azole-treated conidia. Sample

size *n* = 3–4 independent experiments with 21–61 septa per experiment. F-actin rings, labelled with LifeAct-ZtGFP (indicated by LifeAct) in untreated (**h**) and azole-treated (**i**) conidia. Images are contrast inverted; 90°-tilted images provided. Scale bar = 2 μm. See Supplementary Movie 5. **j** Localisation of the putative F-BAR protein ZtImp2-ZtGFP (green, Imp2) at septa in methanol-treated cells (Control) and epoxiconazole-treated cells (Epoxiconazole). The plasma membrane is labelled with mCherry-Sso1 (red; Sso1). Scale bar = 2 μm. See Supplementary Fig. 5 and Supplementary Movie 4. Cells were grown in YG media at 18 °C, 200 rpm, and treated with 0.01 μg ml⁻¹ of the azoles for 24 h. Results shown in (**a**, **d**, **f**) were obtained independently in three experiments and (**e**, **h**, **I**, **j**) in two experiments. Data in (**b**, **c**) did not pass a normality test (Shapiro-Wilk test, all *P* values < 0.0035) and are given as Whiskers' plots with 25/75 percentiles (blue lines), median (red line) and minimum and maximum (whiskers ends); bars in (**g**) represent mean ± SEM; red dots represent independent experiments; statistical analysis in (**b**, **c**) was done using non-parametric Mann-Whitney testing, in (**g**) Student's *t* testing; *= different from control at two-tailed *P* = 0.0203 (**g**); ***= two-tailed *P* = 0.0001 (**g**); ****= two-tailed P < 0.0001 (**b**,**c**). Colour-blind friendly images are available in the Supplementary Fig. 4. All data are provided in the Source Data File.

azole-induced alteration in the lipid composition prevents the putative F-BAR protein ZtImp2 from binding to the PM. This may result in impaired assembly of actin rings and, consequently, incomplete septation.

### Azole treatment hyperpolarises mitochondria and increase ATP levels

Whereas most ergosterol is found in the PM, this sterol is also enriched in various organelle membranes, including and the inner mitochondrial membrane (IMM)[13]. This localisation in mitochondria is of particular interest, as fungal mitochondria are a target for major classes of fungicides[3,31]. To assess the potential effect of azoles on the IMM,

we performed electron microscopy on mitochondria in azole-treated cells. We found that cristae were formed, but they appeared smaller, reduced in number and only rarely formed membrane stacks (Fig. 3a, b, Supplementary Fig. 6a, b; see Supplementary Fig. 7a for control). The IMM contains the enzyme complexes of the respiration chain, required to build the proton-motive force to synthesise ATP[32], and we considered it possible that this oxidative phosphorylation is affected by azoles. To test this, we stained cells with the red-fluorescent membrane potential dye tetramethylrhodamine (TMRM), which accumulates in the mitochondrial matrix in *Z. tritici* due to the polarisation of the IMM[26]. We found weak red-fluorescent signals in cells, treated with the solvent methanol (Fig. 3c, Supplementary Fig. 6c;

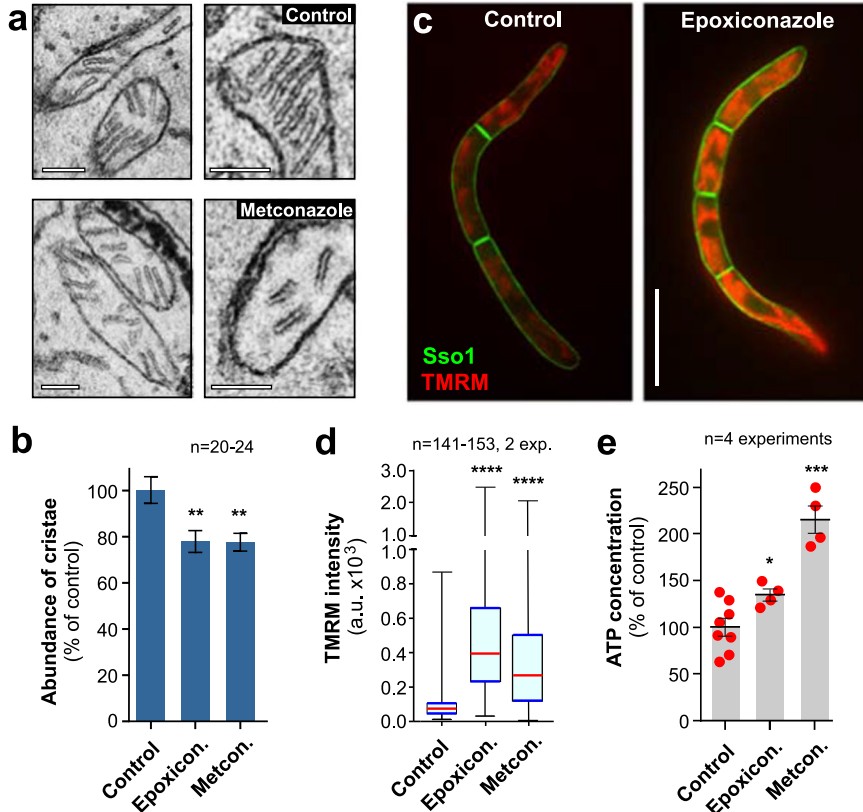

**Fig. 3 | Azoles hyperpolarise mitochondria and increase cellular ATP.**
**a** Ultrastructure of *Z. tritici* mitochondria, treated with 0.01 μg ml⁻¹ metconazole for 24 h. Scale bars = 0.2 μm (left images) and 0.15 μm (right images). See Supplementary Fig. 7a for mitchondria in control cells. **b** Relative abundance of cristae in control and azole-treated mitochondria. Data derived from electron microscopy images. Area of cristaer per area in control cell was set to 100%. Sample size $n = 20$–$24$ from two independent experiments. **c** Mitochondrial membrane potential (red, TMRM) in eGFP-ZtSso1-expressing cells (green, Sso1). Images were acquired using the same microscopic settings and were image processed identically. Scale bar = 10 μm. See Supplementary Fig. 7b. **d** Mitochondrial membrane potential, determined by quatitative TMRM staining, in solvent- (Control), epoxioconazole- (Epoxicon.) or metconazole- (Metcon.) treated cells. Cells were incubated for 24 h at 0.01 μg ml⁻¹ azoles or the solvent. Sample size of 141–153 cells from two independent experiments. **e** Cytplasmic ATP concentration in *Z. tritici* cells, treated with the solvent methanol (Control), epoxiconazole (Epoxicon.) and

metconazole (Metcon.). Sample size $n = 4$ independent experiments. All experiments were done using cells that were treated for 24 h with 0.01 μg ml⁻¹ azoles. Results shown in (**a**, **c**) were obtained independently in two experiments. Data in (**d**) did not pass a normality test (Shapiro-Wilk test, all *P* values < 0.0001) and are given as Whiskers' plots with 25/75 percentiles (blue lines), median (red line) and minimum and maximum (whiskers ends); bars in (**b**, **e**) represent mean ± SEM; and (**e**), with red dots representing values from independent experiments; statistical analysis in (**d**) was done using non-parametric Mann-Whitney testing, and in (**b**, **e**) Student's *t* testing after Welch's correction; *= significant difference to control at two-tailed $P = 0.0114$ (**e**); **= significant difference to control at two-tailed $P = 0.0057$ (**b**, left bar) and $P = 0.0026$ (**b**, right bar); ***= significant difference to control at two-tailed $P = 0.0008$ (**e**); ****= significant difference to control at two-tailed $P < 0.0001$ (**d**). Colour-blind friendly images are available in the Supplementary Fig. 6. All data are provided in the Source Data File.

Control). Treatment with azoles significantly increased TMRM fluorescence (Fig. 3c, d; Supplementary Fig. 6c, 6d, 7b), which is indicative of hyperpolarisation due to increased respiration activity. To test if this affects ATP production, we determined cytoplasmic ATP levels in these cells and found that azole treatment did increase cytoplasmic ATP concentration (Fig. 3e, Supplementary Fig. 6e). This suggests ergosterol depletion in the IMM hyper-polarises mitochondria, which results in a stronger proton-motive force and increased ATP production.

**Azole treatment induces mROS-dependent apoptotic cell death**
Mitochondrial respiration produces reactive oxygen species (mROS), which, when produced in larger quantities, can be harmful to lipids and proteins in the cell[33]. As azoles hyperpolarise the IMM, we asked if azole treatment raises mROS levels. We treated *Z. tritici* with epoxiconazole and metconazole and stained cells with the dye dihydrorhodamine–123 (DHR123). This dye becomes fluorescent after oxidation and was used in *Z. tritici* to detect mROS[26]. Indeed, we found increased DHR123 fluorescence in mitochondria (Fig. 4a, b;

Supplementary Figs. 7c, 8a, 8b,), confirming that mROS production is induced in the presence of azoles.

Elevated mROS activates apoptotic cell death in *Z. tritici*[26], and we asked if epoxiconazole and metconazole induce this type I cell death pathway.

A hallmark of apoptotic cell death is nuclear condensation and fragmentation[34]. We performed electron microscopy studies to test for such phenotype in azole-treated *Z. tritici* cells. Control cells contain single large nuclei (Supplementary Fig. 9a), whereas nuclei in azole-treated cells were smaller and more irregular (Fig. 4c; Supplementary Fig. 8c, 9b). Labelling nuclear DNA with His1-ZtGFP showed that these structures often appeared 'fuzzy', suggesting that they represent nuclear fragments (Fig. 4d, Supplementary Fig. 8d arrowheads in inserts; Supplementary Fig. 9c). This result provided first indications that azoles may induce apoptosis in *Z. tritici*. Next, we tested directly for apoptotic cell death by using two widely-used assays. Firstly, we performed Annexin V-FITC staining, which detects phosphatidylserine, exposed to the outer leaflet of the PM in early apoptotic cells[35], resulting in green-fluorescence (Fig. 4e, Supplementary Fig. 8e, 9d;

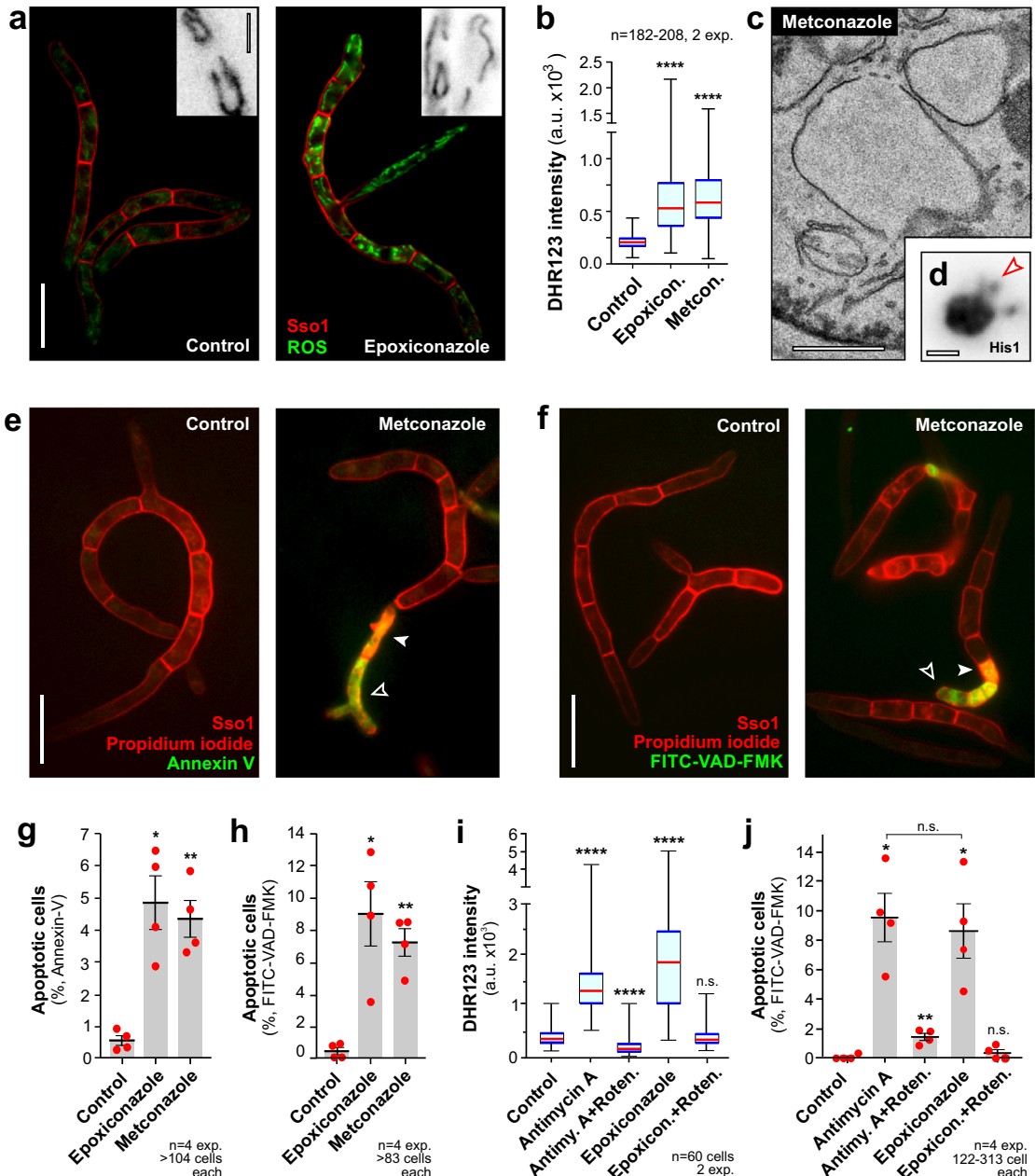

**Fig. 4 | Azoles induce mROS-dependent apoptosis. a** DHR123 staining of mROS (green) in control and epoxiconazole-treated cells. The plasma membrane is labelled withy mCherry-ZtSso1 (red, Sso1); insert shows contrast-inverted images of DHR123-stained mitochondria at higher magnification. Images were acquired using the same microscopic settings and were image processed identically. Scale bars = 10 μm and 3 μm (inset). See also Supplementary Fig. 7c. **b** mROS levels, given as DHR123 fluorescence, in solvent- (Control) and azole-treated cells (Epoxicon., Metcon.). Sample size $n = 182$–208 cells from two independent experiments. **c** Electron micrograph of nuclear fragments in metconazole-treated cells. Scale bars = 0.5 μm. See Supplementary Fig. 9a, 9b. **d** Contrast-inverted image of a nucleus, labelled with DNA-binding His1-ZtGFP, in a metconazole-treated cell. Arowhead indicates 'fuzzy' DNA fragments. Scale bar = 1 μm. Staining of apoptotic cells with Annexin-V (**e**, green) and FITC-VAD-FMK (**f**, green) in control and metconazole-treated cells. Propidium iodide-positive dead cells indicated by filled arrowhead (red, PI). The plasma membrane is labelled with mCherry-ZtSso1 (red, Sso1). Scale bars = 10 μm. See also Supplementary Fig. 9c, 5d. Proportion of apoptotic cells, visualised with Annexin-V (**g**) and FITC-VAD-FMK (**h**) in control and azole-treated cells. Sample size $n = 4$. **i** mROS levels in cells treated with epoxiconazole and inhibitors of mitochondrial respiration complex I (rotenone= Roten.)

and complex III (Antimycin A= Antimy. A). Sample size $n = 60$ from two independent experiments. **j** Number of apoptotic cells after treatment with mROS-inducing Antimycin A (Antimycin A), Antimycin A + rotenone (Antimy.+Roten.), epoxiconazole (Epoxiconazole) and rotenone (Epoxicon.+Roten.). Apoptotic cells were identified by FITC-VAD-FMK staining. Sample size $n = 4$. Cells were grown in YG media at 18 °C and treated with 0.01 μg ml$^{-1}$ azoles for 24 h; for (**i**, **j**), cells were additional treated with 100 μM rotenone or 10 μM antimycin A for 24 h. Results shown in (**a**, **c**, **d**) were obtained independently in two experiments and (**e**, **f**) from independently in 4 experiments. Data in (**b**,**i**) did not pass a normality test (Shapiro-Wilk test, all $P$ values < 0.0128) and are given as Whiskers' plots with 25/75 percentiles (blue lines), median (red line) and minimum and maximum (whiskers ends); bars in (**g**, **h**, **j**) represent mean ± SEM; with red dots representing values from independent experiments; statistical analysis in (**b**, **i**) was done using non-parametric Mann-Whitney testing, and in (**g**, **h**, **j**) Student's $t$ testing; n.s. = non-significant difference to control at two-tailed $P > 0.05$; *= two-tailed $P = 0.0127$ (**g**), $P = 0.0227$ (**h**), $P = 0.0188$ (**j**, Control vs. epoxiconazole) and $P = 0.0103$ (**j**, Control vs. antimycin A); **= two-tailed $P = 0.0052$ (**g**), $P = 0.0029$ (**h**) and $P = 0.0083$ (**j**); ****= two-tailed $P < 0.0001$ (**b**, **i**). Colour-blind friendly images are available in the Supplementary Fig. 8. All data are provided in the Source Data File.

open arrowhead). Secondly, we tested for metacaspase activity in early apoptotic cells using the green-fluorescent FITC-conjugated tripeptide Val-Ala-Asp-fluoro-methyl kerone labelling (FITC-VAD-FMK[36]; Fig. 4f, Supplementary Fig. 8f, 9e, apoptotic cells indicated by open arrowheads). We quantified the number of apoptotic cells using both methods, after excluding dead cells which took up the dyes but also co-stained with the red-fluorescent propidium iodide (Fig. 4e, f, Supplementary Fig. 8e, f; closed arrowheads; overlay results in yellow colour). These experiments revealed that azole-treatment significantly increased the number of apoptotic cells (Fig. 4g, h, Supplementary Fig. 8g, h). Thus, we conclude, that epoxiconazole and metconazole induce type I apoptotic programmed cell death in *Z. tritici*.

Next, we set out to test if azoles induce apoptosis by increasing mROS levels. It was previously shown in *Z. tritici* that mROS generation and subsequent apoptosis can be blocked by inhibition of respiration complex I, using the inhibitor rotenone[26]. In this study, we used antimycin A to increase mROS and trigger apoptotic cell death in *Z. tritici* (Figs. 4i, j, Supplementary Fig. 8i, j). Consistent with the previous report, inhibition of respiration complex I with rotenone prevented mROS production and largely suppressed apoptosis (Fig. 4i, j, Supplementary Fig. 8i, j; Antimy. A+Roten.). Next, we tested epoxiconazole under the same conditions. In control experiments, epoxiconazole induced mROS production and apoptosis (Fig. 4i, j, Supplementary Fig. 8i, j; Epoxiconazole), but rotenone abolished both effects (Fig. 4i, j, Supplementary Fig. 8i, j; Epoxicon.+Roten.). These data suggest that epoxiconazole-induced mROS triggers apoptotic cell death.

## Azole treatment induces macroautophagy

mROS was reported to trigger autophagy in mammalian cells[37]. We therefore asked if azole induce this 'self-eating' pathway in *Z. tritici*. The formation of autophagosomes is a hallmark of autophagy[38]. These autophagic organelles carry the ubiquitin-like protein Atg8, and when fused to green-fluorescent protein was used to monitor autophagy in fungi[39,40], including *Z. tritici*[24]. In control cells, treated with the solvent methanol, few and small eGFP-Atg8-positive autophagocytic organelles were found (Fig. 5a, Supplementary Fig. 10a, arrowheads in 'Control'; images represent a maximum-projection of all focal planes). When cells were treated with $0.01\,\mu g\,ml^{-1}$ epoxiconazole or metconazole for 24 h, this number significantly increased (Fig. 5a, b, Supplementary Fig. 10a, b). We next investigated if mROS initiates autophagy. Antimycin A did induce mROS (see above) but did not increase the number of eGFP-Atg8-positive structures in control cells (Fig. 5c, Supplementary Fig. 10c, Antimycin A). Moreover, rotenone treatment did not prevent epoxiconazole-induced increase in autophagosome numbers (Fig. 5c, Supplementary Fig. 10c, compare 'Epoxicon.+Roten.' and 'Epoxiconazole'). Thus, initiation of autophagy does not involve mROS.

We noticed that azole-treated cells occasionally contained eGFP-Atg8-positive structures that were $>1\,\mu m$ in diameter (Fig. 5d, Supplementary Fig. 10d). These were often 'cup-shaped' (Fig. 5e, Supplementary Fig. 10e) and increased in number with time of treatment (Supplementary Fig. 11a). This morphology is characteristic for macroautophagosomes[39], which could engulf large organelles, such as nuclei, in mammalian cells and fungi[40,41]. To test this notion, we generated a strain that co-expresses eGFP-Atg8 and the genomic DNA marker His1-mCherry (strain IPO323_eGAtg8_His1mCh). Indeed, we found that the large eGFP-Atg8-positive organelles engulf entire nuclei (Fig. 5f, Supplementary Figs. 10f, 11b). We confirmed this finding by electron microscopy (Fig. 5g, Supplementary Fig. 10g; arrowheads highlight a putative autophagosome). This ultrastructural analysis also revealed that the putative autophagosomes were surrounded by multiple membranes (Supplementary Fig. 11c), which is another hallmark of autophagocytic organelles[39]. Moreover, beside nuclei, autophagosomes did contain various cytoplasmic content (Supplementary Fig. 11d). Taken together these results strongly suggest that azoles

induce autophagy, which results in removal of cytoplasm and organelles from the pathogen cell.

Autophagy is considered a programmed cell death pathway (type II cell death[42]). To test if autophagy underpins the fungicidal activity of azoles, we generated *atg4* null mutants in *Z. tritici* (strain IPO323_ΔAtg4). These mutants were neither impaired in growth on agar plates, nor were affected in morphology (Supplementary Fig. 11e, 11f). In the budding yeast, the regulatory protease Atg4 is required for autophagosome formation and Δ*atg4* mutants are impaired in autophagy[43]. We expressed eGFP-Atg8 in IPO323_ΔAtg4 and treated the resulting strain with epoxiconazole. Consistent with a defect in autophagosome biogenesis, we found significantly less autophagic organelles in the mutant when treated with the solvent (Fig. 5h, Supplementary Fig. 10h, compare WT 'Control' with Δ*atg4* 'Control'). Azole treatment of this mutant did no longer induce the formation of small autophagocytotic organelles (Fig. 5h, Supplementary Fig. 10h, $<1\,\mu m$; Δ*atg4* 'Epoxiconazole' not significantly different from WT 'Epoxiconazole'; two-tailed $P = 0.0815$, Mann-Whitney testing) and strongly reduces the appearance of large autophagosomes (Fig. 5i, Supplementary Fig. 10i, $>1\,\mu m$; Δ*atg4* 'Epoxiconazole'). Next, we tested if the inhibition of autophagy in Δ*atg4* mutants altered the fungicidal impact of epoxiconazole in *Z. tritici*. Indeed, we found that Δ*atg4* mutants were strongly reduced in azole-induced mortality (after 4 d treatment from $80.8 \pm 1.9\%$ mortality in control cells to $36.9 \pm 4.0\%$ mortality in Δ*atg4* mutants; both $n = 3$ experiments; Fig. 5j, Supplementary Fig. 10j).

Finally, we asked if starvation-induced autophagy could increase the sensitivity of *Z. tritici* cells for epoxiconazole. To test for this, we grew eGFP-Atg8-expressing *Z. tritici* cells for 5 days in nitrogen-depleted minimal medium. Under these conditions, the number of eGFP-Atg8-labelled autophagic vesicles increased (Supplementary Fig. 11g), confirming that nitrogen starvation induces autophagy in *Z. tritici*. We next treated these cells with $0.01\,\mu g\,ml^{-1}$ epoxiconazole or the solvent alone and determined cell mortality after 24 h. This analysis revealed that starvation itself significantly increased cell death (Supplementary Fig. 11h; 'Control'). However, correcting the data for epoxiconazole-treated cells by this increase still left a significant rise in azole-induced mortality in starved cells (Student's $t$ test; two-tailed $P$ value of 0.0041; Supplementary Fig. 11h; blue bars). These results confirm our finding that induction of type II autophagic cell death contributes to the fungicidal activity of azoles in *Z. tritici*.

## Two cell death pathways contribute to fungicidal activity of azoles

Our results so far suggested that azoles activate two programmed cell death pathways, namely cell death type I (apoptosis) and cell death type II (autophagy). We next tested for the contribution of both cell death pathways in the fungicidal activity of azoles by co-treating Δ*atg4* mutants with epoxiconazole and rotenone, which prevents mROS production and apoptosis (see above). Epoxiconazole-induced mortality was reduced to from $36.9 \pm 4.0\%$ to $12.0 \pm 0.9\%$ ($n = 3$ experiments) when autophagy and apoptosis were inhibited (Fig. 5j, Supplementary Fig. 10j). We still found incomplete septa, suggesting that the effect of azoles on septation is not directly related to both programmed cell death pathways (Supplementary Fig. 11i). We conclude that lethality of azoles in *Z. tritici* is largely due to the activation of two cell death pathways.

We noted that Δ*atg4* mutants still formed a low number of large autophagosomes (Fig. 5h, i, Supplementary Fig. 10h, i), suggesting that autophagy was not completely inhibited in these mutants. Thus, we decided to investigate the role of autophagy in azole-induced mortality using 3-methyladenine (3-MA), a specific inhibitor of autophagosome formation that targets class III PI3 kinase[44]). Co-treating *Z. tritici* cells for 24 h with epoxiconazole and $25\,\mu M$ 3-MA resulted in a significant reduction of small autophagic organelles (Fig. 5k, Supplementary Fig. 10k, $<1\,\mu m$; Mann-Whitney test $P < 0.0001$), although their

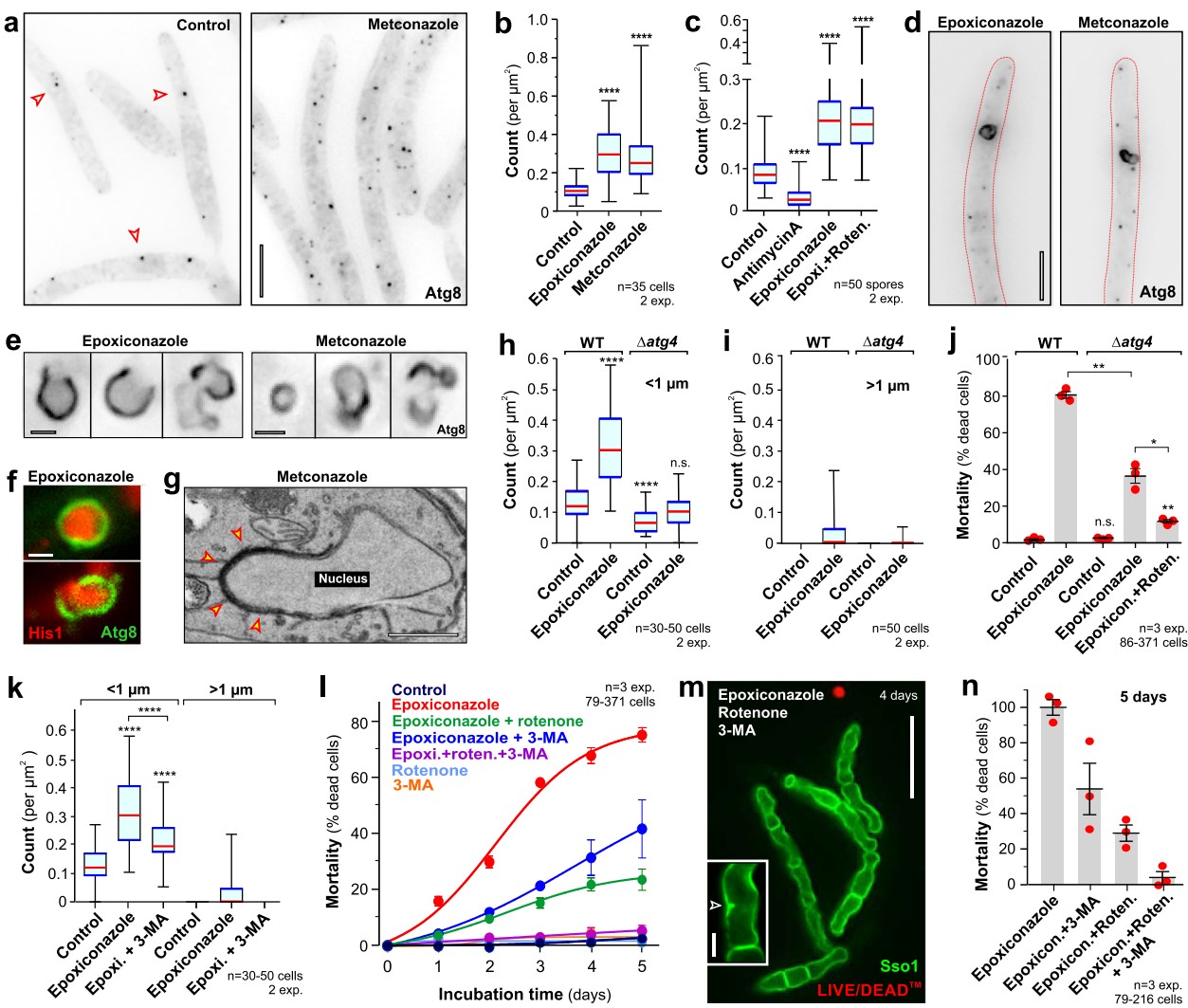

**Fig. 5 | Azoles kill *Z. tritici* cells by inducing apoptosis and autophagy. a** Contrast inverted maximum-projection of autophagosomes, labelled with eGFP-Atg8 (Atg8) in cells treated with solvent (Control) or metconazole; arrowheads indicate individual autophagosomes. Scale bar = 5 μm. **b** Number of autophagosomes in control and azole-treated cells. Sample size *n* = 35 cells from two independent experiments. **c** Number of autophagosomes under increased mROS levels (AntimycinA) and epoxiconazole-treated cells with reduced mROS levels. mROS is not controlling autophagosome numbers. Sample size *n* = 50 cells from two independent experiments. Contrast-inverted maximum projections of large autophagosomes, labelled with eGFP-Atg8 in azole-treated cells. The cell border in (**d**) is indicated by red dotted line. Scale bar = 5 μm (**d**) and 1 μm (**e**). See Supplementary Movie 6. **f** Co-visualisation of eGFP-Atg8 (green, Atg8) and His1-mCherry (red, his1) in living epoxiconazole-treated cells. Scale bar = 1 μm. See Supplementary Fig. 11b. **g** Ultrastructure of a putative large autophagosome (arrowheads), engulfing a nucleus in a metconazole-treated cell. Scale bar = 0.5 μm. See also Supplementary Fig. 11d. Small autophagosomes (<1 μm; **h**) and large autophagosomes (>1 μm; **i**) in wildtype (WT) and Δ*atg4* mutants, treated with solvent (control) and azoles. Sample size *n* = 30–50 cells from two independent experiments. **j** Mortality of epoxiconazole-treated *Z. tritici* cells in wildtype and Δ*atg4* mutant background. Sample size *n* = 3 independent experiments with 86-371 cells. **k** Effect of the autophagy inhibitor 3-MA on epoxiconazole-induced autophagosome formation. Sample size *n* = 30–50 cells from two independent experiments. **l** Mortality of *Z. tritici* cells, treated for 1–5 days with the solvent (Control), epoxiconazole, rotenone or 3-MA and various combinations of these compounds. Sample size *n* = 3 independent experiments with 79–371 cells. **m** eGFP-Sso1-expressing cells (green, Sso1), co-treated with epoxiconazole, 3-MA and rotenone (4 days) and stained with LIVE/DEAD™ dye (red). Inset shows an incomplete septum in the inhibitor-treated cells. Scale bars = 10 μm (overview), 3 μm (inset). **n** Mortality after 5 days treatment with epoxiconazole alone, or in combination with the inhibitors 3-MA and rotenone. Data taken from (**l**) and corrected for background mortality in cultures grown for 5 days in the presence of the solvent (control). Sample size *n* = 3 independent experiments with 79–216 cells. Cells were grown in YG media, 18 °C, 200 rpm and treated 24 h with 0.01 μg ml⁻¹ azoles or as indicated; rotenone was used at 100 μM and 3-MA at 5 μM for 24 h. Results shown in (**a**, **d**, **e**, **f**, **g**) were obtained independently in two experiments and (**m**) from three experiments. Data in (**j**, **l**, **n**) given as mean ± SEM, with red dots representing values from independent experiments; some data sets in (**b**, **c**, **h**, **i**, **k**) did not pass a normality test (Shapiro-Wilk test; *P* < 0.05) and thus all data are given as Whiskers' plots with 25/75 percentiles (blue lines), median (red line) and minimum and maximum (whiskers ends); statistical analysis in (**b**, **c**, **h**, **i**, **k**) used non-parametric Mann-Whitney testing; **j** used Student's *t* testing with Welch correction; data in symbols above whiskers indicate comparison with control; brackets indicate other comparisons; n.s. = non-significant difference two-tailed *P* > 0.05; *= two-tailed *P* = 0.0202 (**j**); **= two-tailed *P* = 0.0026 (**j**, WT 'Epoxiconazole' compared to Δ*atg4* 'Epoxiconazole') and *P* = 0.0069 (**j**, Δ*atg4* 'Epoxicon.+Roten.' versus Δ*atg4* 'Control'); ****= two-tailed *P* < 0.0001 (**b**, **c**, **h**, **k**). Colour-blind friendly images are available in the Supplementary Fig. 10. All data are provided in the Source Data File.

number was still significantly higher than that of control cells (Fig. 5k, Supplementary Fig. 10k, <1 μm; Mann-Whitney test *P* < 0.0001). However, 3-MA prevented the formation of large autophagosomes in epoxiconazole-treated cells (Fig. 5k, Supplementary Fig. 10k, >1 μm;

'Epoxi. + 3-MA'), suggesting that it inhibits autophagy of large organelles, such as nuclei. We next determined the fungicidal activity of epoxiconazole in *Z. tritici* cells, co-treated with the fungicide, 3-MA and rotenone over 5 days. We also determined the mortality of cells,

incubated in 3-MA and rotenone alone, as well as in the presence of the solvent methanol (0.1% [v v⁻¹] methanol; Control) and epoxiconazole alone. These experiments revealed low mortality in cultures exposed to the solvent, as well as rotenone or 3-MA alone (percentage of dead cells after 5 days incubation: control= 4.4%, rotenone= 3.0%, 3-MA = 4.0%; not significantly different, one-side ANOVA, $P$ = 0.2757; Fig. 5l, Supplementary Fig. 10l). On the other hand, 0.01 μg ml⁻¹ epoxiconazole increased mortality to 76.7% over 5 days (Fig. 5l, Supplementary Fig. 10l). When autophagy or apoptosis was inhibited by 3-MA or rotenone, azole-induced mortality significantly dropped (Fig. 5l, Supplementary Fig. 10l). Simultaneously inhibition of autophagy and apoptosis almost abolish the remaining fungicidal activity of epoxiconazole (Fig. 5m, Supplementary Fig. 10m, 'Epoxi. + roten. + 3-MA; mortality after 5 days incubation not significantly different from Control, Student's $t$ test, $P$ = 0.2452), although this treatment strongly affected cell morphology (Fig. 5i, Supplementary Fig. 10i). To estimate the contribution of either programmed cell death pathway, we corrected the mortality of epoxiconazole at 5 days for the number of dead cells in control experiments to remove the 'background mortality' in these *Z. tritici* cultures. The remaining mortality (74.1% of all cells) was due to the activity of epoxiconazole and was set to 100%. When autophagy was suppressed, 53.9% mortality remained, suggesting that mROS and apoptosis account for half of the fungicidal activity of azoles. 28.9% of the cells were killed by epoxiconazole when apoptosis was blocked, which confirms that macroautophagy participates significantly in killing pathogen cells (Fig. 5n, Supplementary Fig. 10n). Blocking both pathways reduced the fungicidal activity of the azole to 3.8% (Fig. 5n, Supplementary Fig. 10n). Thus, very few cells died, suggesting that both programmed cell death pathways underpin almost the entire fungicidal activity of epoxicoinazole. The surviving cells still contained incomplete septa (Fig. 5m, Supplementary Fig. 10m, inset), suggesting that the septation defect is neither a consequence of autophagy nor apoptosis. Taken together, these results strongly suggest that azoles kill fungal pathogens by activating two independent programmed cell death pathways, namely apoptosis and autophagy.

## Azole treatment induces apoptosis and autophagy in pathogen cells *in planta*

Our in vitro experiments focussed on the effect of azoles at low concentrations (0.01 μg ml⁻¹), which approximately resembled the respective MIC for epoxiconazole and metconazole (see above). We next set out to investigate if mROS development, apoptosis and autophagy is also initiated at increased azole concentration (1 μg ml⁻¹). We also tested if the PM integrity is affected under these conditions, as such defect was proposed as the MoA for azoles[14,15]. We found that high azole concentrations did introduce mROS, apoptosis and autophagy (Supplementary Fig. 12a–c). However, at this high concentration, both azoles affect the ion-permeability of the PM, indicated by an increase in depolarised cells (Supplementary Fig. 12d).

We next asked if azoles show a similar MoA in *Z. tritici* cells that are applied to wheat leaves. We firstly investigated the fungicidal potential of epoxiconazole *in planta*. To this end, we sprayed intact wheat plants with epoxiconazole, followed by inoculation with cytoplasmic ZtGFP expressing *Z. tritici* conidia after 24 h. After additional 2 and 5 days, mortality of the pathogen cells on leaf surfaces was assessed, using propidium iodide. We found that most conidia were dead after 2 days exposure to the azole (Fig. 6a, Supplementary Fig. 13a; red/magenta), whereas many hyphae were still alive, suggesting that they are more tolerant of epoxiconazole (Fig. 6a, Supplementary Fig. 13a, arrowheads). However, almost all hyphae were killed on azole-treated leaves at 5 days after inoculation (Fig. 6b, Supplementary Fig. 13b). Thus, azoles sprayed onto wheat leaves kill both, conidia and hyphae.

We next set out to test if azoles induce mROS, apoptosis and autophagy in *Z. tritici* cells *in planta*. To reduce the technical challenges of cell staining *in planta*, we generated 8 reporter strains that express GFP under the control of promoters of genes, known to be transcriptionally up-regulated when (i) ergosterol is depleted, (ii) mROS is produced, (iii) apoptosis is induced, or (iiii) marcroautophagy is triggered (for information of genes and references see Supplementary Table 4; for cloning information see Methods). When tested in liquid culture (in vitro), all *Z. tritici* reporter strains showed a significant increase in cytoplasmic GFP signal intensity (Fig. 6c, Supplementary Fig. 13c, expression of GFP under the promoter of a metacaspase gene shown as example; Fig. 6d, Supplementary Fig. 13d), confirming their suitability for azole MoA detection in *Z. tritici*. We next infected epoxiconazole-sprayed wheat plants with these reporter strains and analysed GFP signal intensity at 2 days post inoculation (2 dpi). This revealed that all markers are induced *in planta* when cells were exposed to the azole (Fig. 6e, f, Supplementary Fig. 13e, f). We conclude that azoles induce two cell death pathways in vitro and *in planta*. We also tested if *Z. tritici* cells on the surface of epoxiconazole-treated plant leaves are depolarised, as this MoA was found at high azole concentrations in vitro (see above; Supplementary Fig. 12d). We did not find a significant increase of DiBAC₄(3)-positive pathogen cells when wheat leaves were pre-treated with 0.01 μg ml⁻¹ or 1 μg ml⁻¹ epoxiconazole (Fig. 6g, Supplementary Fig. 13g). This suggests that the fungicidal activity of azoles *in planta* is due to activation of programmed cell death, rather than disruption of the plasma membrane.

## The MoA of azoles is conserved amongst fungal plant pathogens

So far, our analysis focussed on the wheat pathogen *Z. tritici*. We next asked if the described MoA of azoles also occurs in the rice blast disease fungus *Magnaporthe oryzae*. Firstly, we determined the effect of azoles on the diameter of agar-grown colonies. We found 50% inhibition of growth at -0.12–0.14 μg ml⁻¹ and no hyphal extension at 2–6 μg ml⁻¹ (Supplementary Fig. 2c, 2d). Next, we treated liquid-grown hyphae of *M. oryzae* with various concentrations of epoxiconazole and metconazole and monitored cell death after 24 h, using LIVE/DEAD™ staining.

Cells in the control experiments, treated with the solvent methanol alone, showed -10% 'background' mortality, and additional 10% of the cells were killed in the presence of -10 μg ml⁻¹ (*M. oryzae*; Fig. 7a, Supplementary Fig. 14a). After 4 days at 10 μg ml⁻¹ azoles, -70% all hyphal tip cells were dead (Supplementary Fig. 15). Thus, metconazole and epoxiconazole are -1000 times less fungicidal in *M. oryzae* compared to *Z. tritici* (-70% dead after 4 days at 0.01 μg ml⁻¹, see above). All subsequent experiments were done after 24 h incubation in 10 μg ml⁻¹, which represents -1.7–5 times the MIC of azoles in agar plate growth assays (Supplementary Fig. 2c, d).

We firstly set out to test the effect of azole treatment on septation within the tip cell of *M. oryzae* hyphae, expressing the PM marker Sso1_eGFP[45] (kindly provided by Prof. N. Talbot). This revealed that azole-treated hyphae increased the numbers of septa within the apical 50 μm of the tip cell (Fig. 7b, Supplementary Fig. 14b). However, many of the Sso1-eGFP-labelled septa appeared incomplete and disorganised (Fig. 7c, d, Supplementary Fig. 14c, d; Supplementary Movie 7). Ultrastructural studies confirmed that azoles induced incomplete and altered septae that usually lacked Woronin bodies (Fig. 7e, Supplementary Fig. 14e, 16). Thus, we conclude that azoles affect septation in *M. oryzae*, again confirming the MoA we report in *Z. tritici*.

Next, we set out to investigate if azole treatment has an impact on mitochondria in *M. oryzae* hyphae. Indeed, azole treatment resulted in hyperpolarisation (Fig. 7f, Supplementary Fig. 14f) and induced the formation of mROS (Fig. 7g, Supplementary Fig. 14g). Similar to *Z. tritici*, mROS formation was accompanied by induction of apoptotic cell death (Fig. 7h, Supplementary Fig. 14h; monitored using FITC-VAD-FMK staining). We also tested for the induction of autophagy by azoles, using a *M. oryzae* strain that contained GFPAtg8-labelled

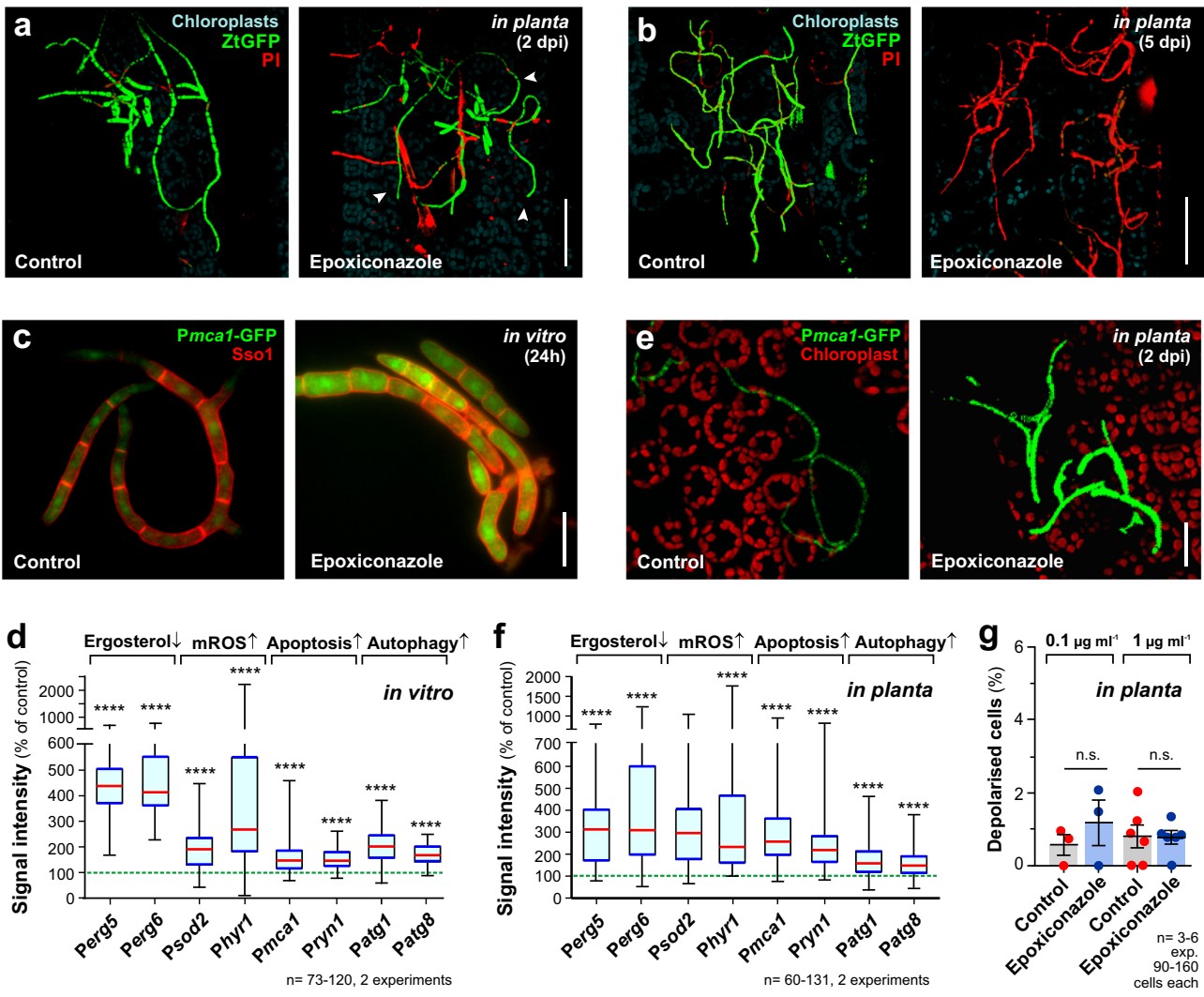

**Fig. 6 | Activity of epoxiconazole in *Z. tritici* during early plant infection.** Epoxiconazole-induced mortality of *Z. tritici*, expressing cytoplasmic codo-optimised GFP (green, ZtGFP) on the plant leaf surface. Wheat leaves were pre-treated with the solvent methanol (Control) or 0.1 μg ml⁻¹ epoxiconazole, followed by inoculation with pathogen conidia 24 h later. LIVE/DEAD staining using propidium iodide was performed 2 days (**a**) and 5 days (**b**) later. Dead cells appear red, green cells are alive. Arrowheads indicate hyphae. Scale bars = 50 μm. **c** Expression of ZtGFP under the control the promoter of the metacaspase gene *mca1* (green, P*mca1*-ZtGFP) in liquid culture; the PM is labelled with mCherry-ZtSso1 (red, Sso1). Expression of *mca1* is induced in the presence of epoxiconazole, resulting in green-fluorescent cytoplasm. Images were acquired using the same microscopic settings and were image processed identically. Scale bars = 10 μm. **d** Quantitative analysis of cytoplasmic green fluorescence in liquid culture-grown marker strains exposed to epoxiconazole. The marker strains respond to reduced ergosterol levels (P*erg5*, P*erg6*), increased mROS (P*sod2*, P*hyr1*), apoptosis (P*mca1*, P*ryn1*) or autophagy (P*atg1*, P*atg8*) by inducing ZtGFP expression. The green-dotted line represents the median of fluorescent intensity in cells exposed to the solvent alone; this value was set to 100%. Sample size *n* = 73–120 cells from two independent experiments. For further information on marker genes see Supplementary Table 4. **e** Expression of ZtGFP under the control the promoter of the metacaspase gene *mca1* (green, P*mca1*-ZtGFP); chloroplast autofluorescence is shown in red. Expression of *mca1* is induced in the presence of epoxiconazole, resulting in green-fluorescent

cytoplasm. Images were acquired using the same microscopic settings and were image processed identically. Scale bars = 20 μm. **f** Quantitative analysis of cyto-plasmic green-fluorescence in marker strains that rested on leaf surfaces for 2 days. The leaves were pre-sprayed with epoxiconazole or the solvent methanol (control). The green-dotted line represents the median of fluorescent intensity in control cells; this value was set to 100%. Sample size *n* = 60–131 cells from two independent experiments. For further information on marker genes see Supplementary Table 4. **g** Relative number of depolarised *Z. tritici* cells on plant surfaces. Propidium iodide-positive cells (=dead cells) were excluded from the analysis. Sample size *n* = 3–6 independent experiments with 60–131 cells each. For in vitro experiments (**c,d**), cells were treated for 24 h with 0.01 μg ml⁻¹ epoxiconazole; for *in planta* experiments, 12 day-old wheat plants were sprayed with 0.1 μg ml⁻¹ (**a, b, e, f, i, g**) or 1 μg ml⁻¹ (**g**) epoxiconazole and imaged at 1 dpi (**g**), 2 dpi (**a, e, f**) or 5 dpi (**b**). Results shown in (**a, b, c, e**) were obtained independently in two experiments. Data in (**g**) given as mean ± SEM, with dots representing values from independent experiments; some data sets in (**d, f**) did not pass a normality test (Shapiro-Wilk test; *P* < 0.05) and thus all data are given as Whiskers' plots with 25/75 percentiles (blue lines), median (red line) and minimum and maximum (whiskers ends). Statistical analysis in (**e, f**) used non-parametric Mann-Whitney testing and in (**g**) Student's *t* testing; n.s.= not significantly different at two-tailed *P* > 0.05; ****= statistical dif-ference to control at two-tailed *P* value < 0.0001. Colour-blind friendly images are available in the Supplementary Fig. 13. All data are provided in the Source Data File.

autophagosomes[46]. Treatment with 10 μg ml⁻¹ azoles for 24 h sig-nificantly increased the number of autophagic structures (Fig. 7i, Supplementary Fig. 14i). Again, large autophagosomes were found (Fig. 7j, Supplementary Fig. 14j; arrowheads), which contained mem-branous organelles (Fig. 7k, Supplementary Fig. 14k, left panel) and

occasionally engulfed entire nuclei (Fig. 7k, Supplementary Fig. 14k, right panels). Thus, we conclude that azoles induce autophagy in *M. oryzae*.

As azoles induced apoptosis and autophagy in *M. oryzae*, we asked if inhibition of both cell death pathways abolish the fungitoxic activity

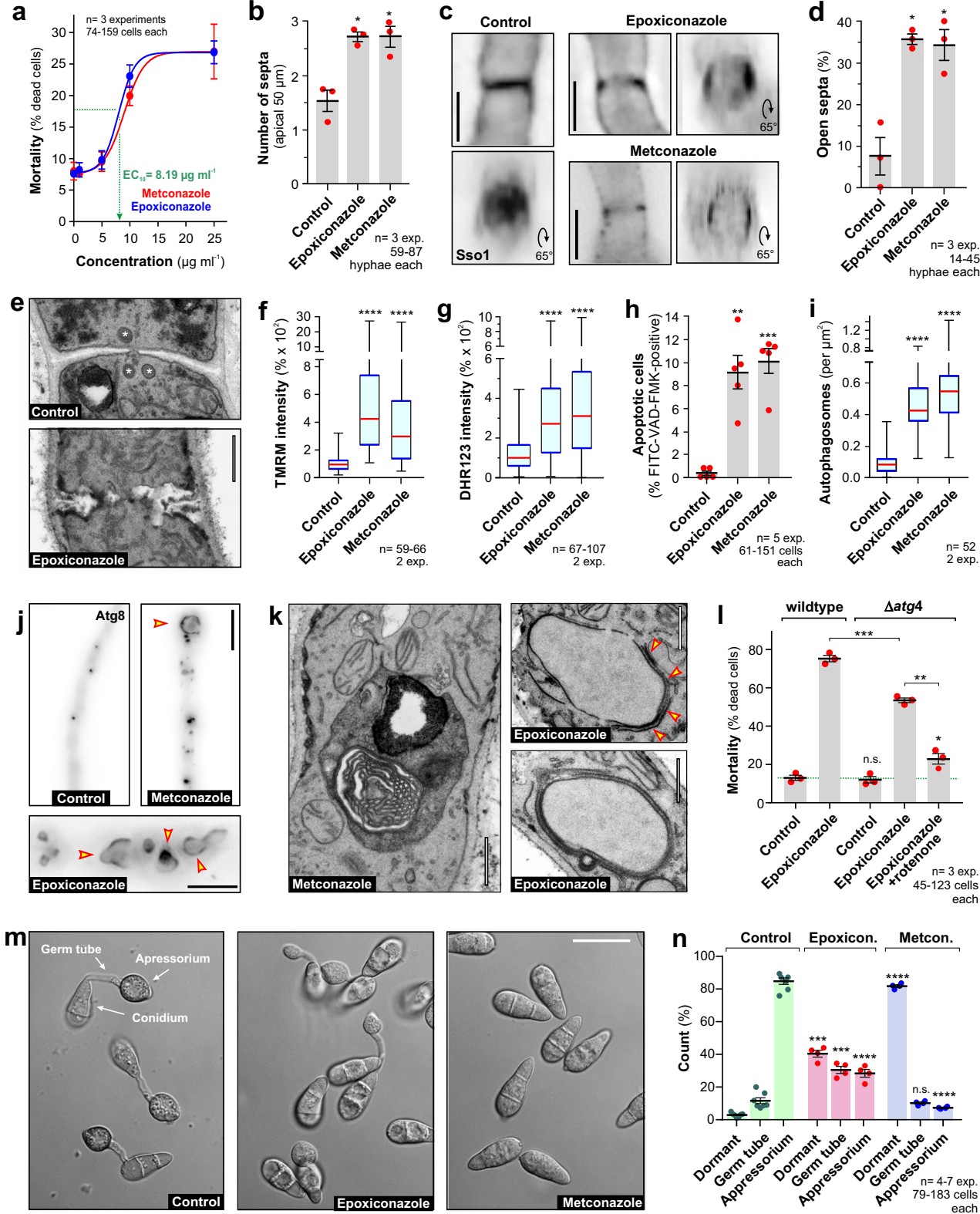

of epoxiconazole in this pathogen, too. In analogy to the work in *Z. tritici*, we tested for epoxiconazole-induced cell mortality in an Δ*atg4* mutant of *M. oryzae*[46] (kindly provided by Prof. N. Talbot). We found that Δ*atg4* mutants are significantly less sensitive to epoxiconazole than wildtype cells and mortality declined by 33.4% (Fig. 7l, Supplementary Fig. 14l; Student's *t* test, *P* = 0.0005). Co-treatment of Δ*atg4* mutants with epoxiconazole and rotenone further reduced the lethality of the azole in *M. oryzae* by 51%, leaving ~10% mortality over the

untreated control cells after 4 days of incubation (Fig. 7l, Supplementary Fig. 14l, green dotted line indicates mortality in control cultures). These results demonstrate that azoles kill *Z. tritici* and *M. oryzae* by activating two cell death pathways.

At the onset of rice infection, *M. oryzae* conidia form a germ tube that differentiates into an appressorium[47], and we finally investigated if azoles inhibit this crucial developmental step, using a previously established in vitro assay[26]. We found that significantly less germ tubes

**Fig. 7 | The rice blast pathogen *M. oryzae* shares the azole MoA with *Z. tritici*.**
**a** Mortality of hyphal *M. oryzae* cells, treated with azoles for 24 h. The effective concentration at 10% increase of mortality over background mortality in control sample (EC$_{10}$) is indicated in green. Sample size $n = 3$ independent experiments with 74–159 cells each. See also Supplementary Fig. 15 for mortality at higher concentrations and Supplementary Fig. 2c, d for growth inhibition on agar plates.
**b** Number of septa, labelled with *M. oryzae* Sso1, fused to eGFP (Sso1-eGFP) in the hyphal apical 50 μm, treated with solvent (Control) and azoles. Sample size $n = 3$ independent experiments with 59-87 hyphae each. **c** Sso1-eGFP-labelled septa (Sso1) in solvent- (control) and azole-treated hyphae. Images are contrast inverted. Tilting 3D reconstructions are given (65°). Scale bars = 3 μm. See Supplementary Movie 7. **d** Incomplete septae in hyphae, treated with solvent (Control) and azoles. Sample size $n = 3$ independent experiments with 14-45 hyphae each.
**e** Ultrastructure of septa in solvent- (control) and epoxiconazole-treated hyphae. Woronin bodies indicated by asterisks. Scale bar = 0.5 μm. See also Supplementary Fig. 16. **f** IMM potential, given as TMRM fluorescence, in hyphae treated with solvent (Control) and azoles. Sample size $n = 59$–66 cells from two independent experiments. **g** mROS levels, given as DHR–123 fluorescence, in hyphae treated with solvent (Control) and azoles. Sample size $n = 67$–107 cells from two independent experiments. **h** Apoptosis, shown as relative number of FITC-VAD-FMK-stained hyphae after 24 h treatment with solvent (Control) and azoles. Sample size $n = 5$ independent experiments with 61–1581 cells each. **i** eGFP-Atg8-labelled autophagosomes in hyphae, treated with solvent (Control) and azoles. Sample size n = 52 cells from two independent experiments. **j** Autophagosomes, labelled with green-fluorescent *M. oryzae* Atg8 (Atg8) in hyphae that were treated with solvent (Control) and azoles. Arrowheads indicate large autophagosomes. Scale bars= 5 μm (upper panels) and 3 μm (lower panel). Images contrast-inverted. **k** Ultrastructure of autophagosomes in azole-treated hyphae. Arrowheads indicate phagosome surrounding a nucleus. Scale bar = 0.5 μm. **l** Mortality of wildtype and Δ*atg4* mutant hyphae, treated with solvent (Control) and azoles for 4 days. Sample size $n = 3$ independent experiments with 45–123 cells each. Germination and appressorium formation after 4–5 h in the presence of 10 μg ml$^{-1}$ epoxiconazole or metconazole. Sample size $n = 4$–7 independent experiments with 79–1831 cells each (**n**). Scale bar in (**m**) = 20 μm. Cells grown in CM-Glucose, 25 °C, 100 rpm and treated for 24 h with 10 μg ml$^{-1}$ azoles. Results shown in (**c**) were obtained independently in three experiments, (**e, j, k**) were obtained independently in 2 experiments and (**m**) were obtained independently in 4-7 experiments. Data sets in (**f, g, i**) did not pass normality testing (Shapiro-Wilk test, $P < 0.05$) and are given as Whiskers' plots with 25/75 percentiles (blue lines), median (red line) and minimum and maximum (whiskers ends); bars in (**a, b, d, h, l, n**) represent mean ± SEM, dots indicate independent experiments; statistical analysis in (**f, g, i**) used non-parametric Mann-Whitney testing, in (**b, d, h, l, n**) Student's t-testing with Welch correction; n.s.= non-significant difference at two-tailed $P > 0.05$; *= significant difference to control at two-tailed $P = 0.0154$ (**b**, Epoxiconazole), 0.0126 (**b**, Metconazole), 0.0187 **d**, Epoxiconazole), 0.0114 (**d**, Metconazole) and 0.0373 (**l**); **= significant difference to control at two-tailed $P = 0.0036$ (**h**) and 0.0029 (**l**); ***= significant difference to control at two-tailed $P = 0.0007$ (**h**), 0.0007 (**l**), 0.0002 (**n**, Dormant) and 0.0003 (**n**, Germ tube);****= two-tailed P < 0.0001 (**f, g, i, n**). Colour-blind friendly images are available in the Supplementary Fig. 14. All data are provided in the Source Data File.

and appressoria are formed in the presence of 10 μg ml$^{-1}$ azoles (Fig. 7m, n, Supplementary Fig. 14m, n). However, metconazole was more effective than epoxiconazole at this sub-lethal concentration. Thus, both azoles are expected to differ in their potential to protect plants against rice blast disease.

**Ergosterol biosynthesis inhibitors share a common MoA**
Azoles are targeting 14-α-demethylase, which catalyses modification of lanosterol and eburicol in the ergosterol biosynthesis (Supplementary Fig. 17). This results in (i) depletion of ergosterol in membranes and (ii) the accumulation of methyl-sterols[48] that are thought to be responsible for azole fungitoxicity[7]. Several ergosterol inhibitors are known that will deplete ergosterol, yet they inhibit different enzymes and thus will not result in accumulation of the same intermediate (Fig. 8a, Supplementary Fig. 18a). We tested the cellular response to the allyamide fungicide terbinafine[49] and the thiocarbamate tolfanate[50], which both inhibit squalene epoxidase, thereby preventing ergosterol biosynthesis without producing methyl-sterols (Fig. 8a, Supplementary Figs. 17, 18a). We also included the morpholine fenpropiomorph, known to inhibit later steps in the ergosterol formation[51] (Fig. 8a, Supplementary Fig. 18a) and fluconazole, an azole widely used to control human pathogenic fungi[52]. We found that all ergosterol synthesis inhibitors impair septation (Fig. 8b, Supplementary Fig. 18b), hyperpolarize mitochondria (Fig. 8c, Supplementary Fig. 18c), elevate mROS levels (Fig. 8d, Supplementary Fig. 18d), induce apoptosis (Fig. 8e, Supplementary Fig. 18e), and stimulate autophagy (Fig. 8f, Supplementary Fig. 18f; green dotted line in Fig. 8b-f and Supplementary Fig. 18b–f indicates average in control). However, the extend of each effect differed between the different fungicides used (Kruskal-Wallis testing Fig. 8c, d, f, Supplementary Fig.18c, d, f). Thus, all inhibitors share a common MoA in *Z. tritici*, which is independent of their specific enzymatic target in the ergosterol biosynthesis. This result supports the notion that the observed cellular defects are a consequence of ergosterol depletion in membranes rather than accumulation of a specific metabolic intermediate.

We finally tested if supply of external ergosterol could rescue the cellular defects, caused by epoxiconazole treatment. Such application of ergosterol was previously shown to enable growth of sterol auxotrophic yeast cells[53]. We found that 5 μg ml$^{-1}$ exogenous ergosterol largely rescued the septation defect and significantly reduced the number of autophagosomes (Supplementary Fig. 19a, b). However, hyperpolarisation of mitochondria was only partially lowered (Supplementary Fig. 19c) and, consequently, neither mROS development nor apoptosis was suppressed (Supplementary Fig. 19d, e). It is currently unclear if the partial rescue of the mitochondria-related defects is due to incomplete uptake of externally-applied ergosterol into the IMM.

## Discussion
Sterols are a major compound of the eukaryotic PM, making up to 20–40% of the lipids in the PM[54]. In fungi, the main sterol ergosterol has important roles in controlling membrane fluidity[55,56] and supporting the activity of PM enzymes[57]. Consequently, depletion of ergosterol is widely thought to compromise the integrity of the PM[14,15], and it was suggested that this underpins the fungicidal activity of certain azole compounds[58]. In this study, we use a fluorescent PM syntaxin GFP-Sso1[24] and show reduces membrane fluidity in azole-treated *Z. tritici* cells, a result that is consistent with findings in *C. glabrata*[59]. We also report that azoles at low concentration (0.01 μg ml$^{-1}$) did not cause impaired PM integrity in cells grown in liquid culture. However, at higher concentration (1 μg ml$^{-1}$), a significant increase of depolarised cells was found, suggesting increased permeability of the plasma membrane. This effect was not seen in pathogen cells that were resting on epoxiconazole-treated wheat leaves. Thus, we consider it likely that this MoA of azole is restricted to treatment in liquid culture and is not of major relevance for the fungicidal activity of azoles in plants.

We do report here that azole treatment induces incomplete septation in *Z. tritici* and *M. oryzae*. Septum formation in fungi depends on the assembly of a contractile actin/myosin ring[28]. We visualised F-actin at sites of septation using the marker protein LifeAct-ZtGFP[24] and show that actin rings are often incomplete or disorganised. This result is reminiscent of findings in *S. pombe*, where altered ergosterol levels in the PM affect assembly of the cytokinetic actin ring[60]. In the fission yeast, the actin ring is anchored to the PM via the F-BAR domain proteins Cdc15 and Imp2[29]. We visualised a fluorescent homologue of Imp2/Cdc15 and found that this protein is misplaced in azole-treated cells. This suggests that that changes in PM lipid composition interferes with the localisation of this anchor protein. This, in turn, affects

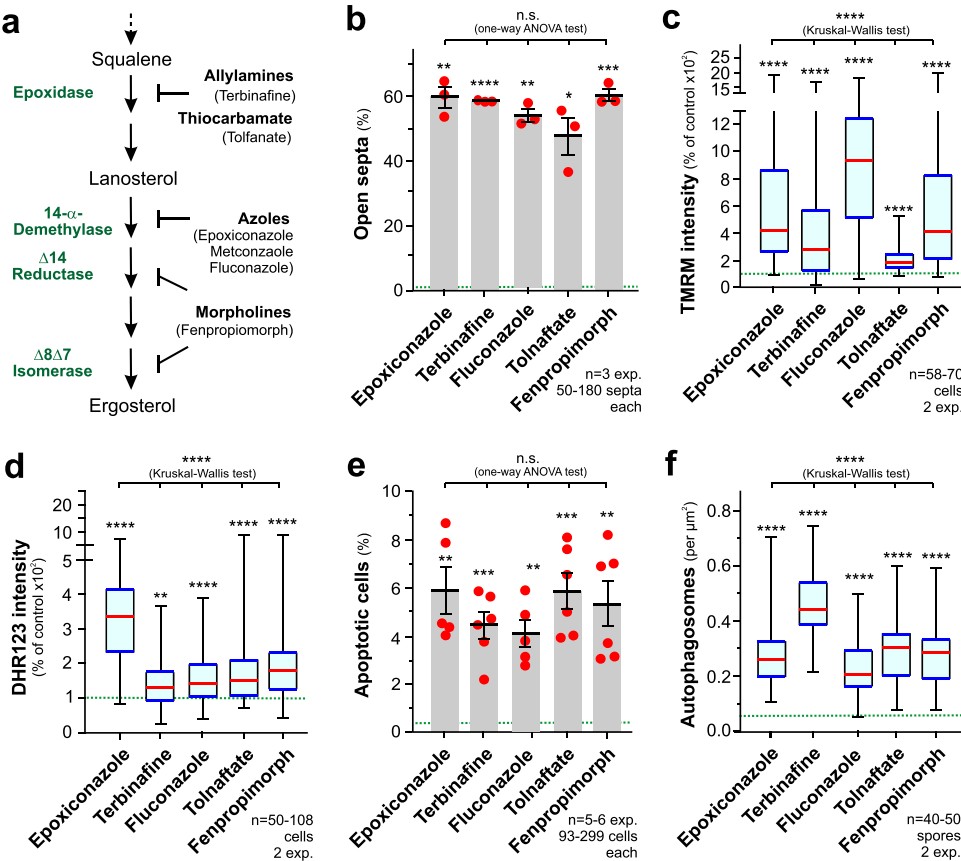

**Fig. 8 | Ergosterol biosynthesis inhibitors have a common MoA. a** Molecular targets of inhibitors of the ergosterol biosynthesis pathway. Enzymes are shown in green; names of inhibitors, used in this study, are provided in parenthesis. See Supplementary Fig. 17. **b** Incomplete septation after treatment with ergosterol biosynthesis inhibitors. Sample size $n = 3$ independent experiments with 50−180 cells each. **c** IMM potential, given as TMRM fluorescence, in cells treated with ergosterol biosynthesis inhibitors. Sample size $n = 58−70$ cells from two independent experiments. **d** mROS, given as DHR−123 fluorescence, in cells treated with ergosterol biosynthesis inhibitors. Sample size $n = 50−108$ cells from two independent experiments. **e** Induction of apoptosis, shown as relative number of FITC-VAD-FMK-stained cells after 24 h, in cells treated with ergosterol biosynthesis inhibitors. Sample size $n = 5−6$ independent experiments with 93-299 cells each. **f** Induction of autophagy, shown as the number eGFP-Atg8-labelled autophagosomes in 1 μm² of a maximum projection of a focal plane in cells treated with ergosterol biosynthesis inhibitors. Sample size n = 40−50 spores from 2 independent experiments. For all experiments cells were grown in YG media at 18 °C with 200 rpm and treated for 24 h with 0.01 μg ml⁻¹ (Epoxiconazole), 5 μg ml⁻¹ (Terbinafine), 50 μg ml⁻¹ (Tolnaftate), 15 μg ml⁻¹ (Fluconazole)and 5 μg ml⁻¹

(Fenmropiomorph); control experiments contained the corresponding amount of the solvent. Most data sets in (**c**, **d**, **f**) did not pass a normality test (Shapiro-Wilk test, $P < 0.05$) and are given as Whiskers' plots with 25/75 percentiles (blue lines), median (red line) and minimum and maximum (whiskers ends); bars in (**b**, **e**) represent mean ± SEM; red dots represent independent experiments; statistical comparison with control in (**c,d,f**) used non-parametric Mann-Whitney testing, and in (**b**, **e**) Student's $t$ testing with Welch correction; multiple data sets were compared using one-way ANOVA (**b**, **e**) or non-parametric Kruskal-Wallis testing (**c**, **d**, **f**); testing results for comparison to control are: n.s.: non-significant difference at one-side $P > 0.05$; *: two-tailed $P$ values of 0.0140 (**b**); **: two-tailed $P$ values of 0.0026 (**b**, epoxiconazole), 0.0012 (**b**, fluconazole) 0.0045 (**d**), 0.0033 (**e**, epoxiconazole), 0.0021 (**e**, fluconazole) and 0.0029 (**e**, fenpropimorph); ***: two-tailed $P$ values of 0.0009 (**b**), 0.0004 (**e**, terbinafine), and 0.0005 (**e**, tolfonate); ****: two-tailed $P$ values < 0.0001 (**b**, **c**, **d**, **f**); testing results for multiple comparisons are: n.s.: non-significant difference at $P = 0.0825$ (**b**) and 0.6983 (**e**); ****= $P$ value < 0.0001 (**c**, **d**, **f**). Colour-blind friendly images are available in the Supplementary Fig. 18. All data are provided in the Source Data File.

assembly and function of the cytokinetic actin ring and causes the observed incomplete septation. While this MoA of azoles in *Z. tritici* is most obvious, it appears not to underpin the observed fungicidal activity of azoles in this pathogen. This conclusion is supported by our finding that septation defects were still found in surviving cells, treated with epoxiconazole and inhibitors of both programmed cell death pathways. Thus, while the septation defect is indicative of alterations in PM sterol content, it does not explain the fungitoxicity of azoles.

Azoles inhibit the synthesis of ergosterol, thereby depleting the sterol in membranes. Studies in *S. cerevisiae* have shown that ergosterol is enriched in the PM[13] but is also found in organelle membranes, such as the IMM[61]. In particular, the IMM is of interest as it contains the complexes of the respiration chain, involved in cellular ATP production during oxidative phosphorylation[32]. Electron transport through the respiration chain requires diffusion-based interaction of the respiration complexes within supercomplexes[62]. As reduced

ergosterol content is expected to change the fluidity of the IMM, we considered it likely that azole-treatment affect this interaction, thereby affecting electron transport through the respiration chain. Indeed, we report here that epoxiconazole and metconazole treatment hyperpolarised the IMM in *Z. tritici* and *M. oryzae*. While under normal conditions, increase cellular ATP is harmless[63], hyperactive mitochondria produce high amounts of mROS, which can oxidise vital proteins and lipids in the IMM[33]. Azole-induced generation of toxic mROS has been reported in human pathogenic fungi, including *A. fumigatus*[18], *Candida albicans*[17] and *Cryptococcus gattii*[64] and it was shown that mROS generation correlates with hyperactivation of mitochondrial respiration and fungicidal activity of the drug[22]. These results suggest that increased respiration activity and ROS production underpin the mortality of azoles in fungi.

We show here that azole treatment results in nuclear fragmentation, surface-exposure of phosphatidylserine (Annexin V staining) and

activation of a caspase activity (FITC-VAD-FMK staining). These cellular responses are consistent with the induction of apoptosis[34–36]. We show that this cell death pathway is induced by mROS, which confirms previous results in *Z. tritici*[26] and higher eukaryotes[65], suggesting that azoles are fungicidal due to their effect on mitochondria, which causes this 'suicide' type I cell death programme. We also report here that azoles induce formation of large autophagosomes that engulf of nuclei and other organelles. Removing nuclei from the cell is expected to be lethal and autophagy is defined as a type II programmed cell death pathway[42]. Indeed, in *Z. tritici* deletion of *atg4* or treatment with the autophagy inhibitor 3-MA prevented formation large phagophore and reduced epoxiconazole-induced mortality to ~50%. When mROS formation and apoptosis and apoptotic cell death were inhibited by 3-MA and rotenone epoxiconazole was no longer fungicidal in *Z. tritici*. Thus, we conclude that autophagy and apoptosis contribute to the fungicidal activity of azoles in both plant pathogens.

In mammalian cells, mROS was reported to induce autophagy[37]. In *Z. tritici*, mROS does not promote autophagy, making it unlikely that the azole-induced defect in mitochondria initiate this pathway. But how does inhibition of the ergosterol pathway promote autophagy? Autophagosome biogenesis is triggered by a broad range of signals, which include DNA damage, amino acid shortage or various cellular stresses[66]. Reduced levels of ergosterol affects PM targeting of amino acid permeases[67], expected to reduce amino acid uptake, and we also show here that azoles cause DNA damage in *Z. tritici*. Thus, autophagy might be initiated by such internal stimuli. Alternatively, reduced ergosterol levels could directly promote autophagosome formation. In mammalian cells, depletion of this cholesterol promotes autophagy initiation, and it was suggested that this results from changes in the fluidity of early autophagosomal membranes[68]. However, understanding the reason for autophagy initiation by azoles requires future investigations.

In this study, we investigate the effect of epoxiconazole and metconazole on the Septoria wheat blotch fungus *Z. tritici* and the rice blast fungus *M. oryzae*. Both fungi range amongst the most important fungal pathogens[23], and the azoles chosen are in use to control fungal disease in the field[69]. In both pathogens, azoles induce septation defects and initiate generation of mROS, which triggers apoptosis but is also harmful on its own[33] (Fig. 9, Supplementary Fig. 20). Our in vitro results show that >50% of the fungicidal activity of epoxiconazole in *Z. tritici* is due to mROS formation and apoptosis. Moreover, azole treatment initiates autophagy, which we show is responsible for ~30% of the fungicidal activity of epoxiconazole. We also report this MoA for fluconazole, an azole commonly used in clinical applications[53]. Thus, we consider it most likely that mROS formation and initiation of apoptosis, as well as autophagy underpins the fungicidal activity of azoles in general.

We also show that an allylamine, a thiocarbamate and a morpholine share this MoA with azoles. These fungicides interfere with various enzymes in the ergosterol biosynthesis pathway (Supplementary Fig. 17), and their inhibition is expected result in accumulation of different intermediate molecules. However, the impact of these ergosterol synthesis inhibitors on the pathogen is very similar. This argues that ergosterol depletion, rather than the proposed accumulation of a toxic sterol intermediate[7], is responsible for the observed impact in the pathogen cell. Finally, we show that mROS development, apoptosis and autophagy are also triggered *in planta*. This suggests that programmed cell death pathways underpin the protective activity of azoles in the field. In needs to be noted that field applications of azoles can use very high concentrations[70]. Thus, we cannot rule out that other, so-far undetected physiological effects in the pathogen support the anti-fungal activity of azoles in crops. In summary, our study provides novel insight into the MoA of azoles and other inhibitors of the ergosterol biosynthesis pathway. In addition, we show for the first time that autophagy can be a target for antifungal compounds.

This knowledge may help developing new strategies to control fungal pathogens.

## Methods

### Fungal strains
The *Z. tritici* wild type strain IPO323 (cbs 115943) was obtained from the Fungal Biodiversity Centre, Utrecht, Netherlands. The strains IPO323, IPO323_His1-ZtG_mCh-Sso1, IPO323_eGSso1, IPO323_Lifeact-ZtG, IPO323_eGAtg8, IPO323_ZtG, IPO323_mChSsoI were described previously (see Supplementary Table 1 for literature references). The *M. oryzae* wild type strain Guy11 (FGSC 9462) as well as strains Guy11_S-so1_GFP, Guy11_GFP-MoATG8 and Guy11_ΔAtg4 were provided by Prof. N. Talbot, Sainsbury Laboratory, Norwich, UK. For genotype and references of all strains see Supplementary Table 1.

### Bacteria
Plasmids were propagated in *Escherichia coli* strain DH5α (Thermo Fisher Scientific, UK). *A. tumefaciens*-mediated transformation of *Z. tritici* used strain EHA105 (GoldBio, St Louis, USA). Bacteria were grown in double-strength yeast extract/tryptone DYT medium (tryptone, 16 g l⁻¹; yeast extract, 10 g l⁻¹; NaCl, 5 g l⁻¹; with 20 g l⁻¹ agar in plates) at 37 and 28 °C, respectively.

### Fungal growth conditions
All fungal strains of *Z. tritici* were maintained as NSY glycerol stocks (nutrient broth, 8 g l⁻¹; yeast extract, 1 g l⁻¹; sucrose, 5 g l⁻¹; glycerol, 700 ml l⁻¹), stored at −80 °C. Spores of *M. oryzae* were stored on filter paper at −20 °C. From these stocks, *Z. tritici* strains were grown on solid YPD agar (yeast extract, 10 g l⁻¹; mycological peptone, 20 g l⁻¹; glucose, 20 g l⁻¹; agar, 20 g l⁻¹) at 18 °C for 4-5 days. Subsequently, cells were inoculated in YG liquid medium (yeast extract, 10 g l⁻¹; glucose, 30 g l⁻¹) and grown for 48 h at 18 °C and 200 rpm, followed by experimentation. For nitrogen starvation experiments, cells were grown for 5 days in minimal medium [per 1 l: 50 ml l⁻¹ of salt solution (KCl, 10.4 g l⁻¹; MgSO₄·7 H₂O, 10.4 g l⁻¹; KH₂PO₄, 30.4 g l⁻¹), 1 ml l⁻¹ trace elements (ZnSO₄·7H₂O, 2.2 g l⁻¹; H₃BO₃, 1.1 g l⁻¹; MnCl₂·4H₂O, 0.5 g l⁻¹; FeSO₄·7H₂O, 0.5 g l⁻¹; CoCl₂·5 H₂O, 0.16 g l⁻¹; CuSO₄·5 H₂O, 0.16 g l⁻¹; (NH₄)₆Mo₇O24·4 H₂O, 0.11 g l⁻¹; Na₄ EDTA, 5 g l⁻¹) and glucose, 10 g l⁻¹], supplemented with or without NaNO₃ (6 g l⁻¹). Subsequently, cells were inoculated with 0.01 µg ml⁻¹ epoxiconazole and grown for 24 h at 18 °C and 200 rpm, followed by LIVED/DEAD™ staining and microscopic analysis.

*M. oryzae* cells were stored at −20 °C on dried Whatman filter paper squares. Cells were grown from those fungal filter stocks by placing one square onto complete medium (glucose, 10 g l⁻¹; peptone, 2 g l⁻¹; yeast extract, 1 g l⁻¹; casamino acids, 1 g l⁻¹; EDTA, 50 mg l⁻¹; zinc sulphate heptahydrate, 22 mg l⁻¹; boric acid, 11 mg l⁻¹; manganese (II) chloride tetrahydrate, 5 mg l⁻¹; iron (II) sulphate heptahydrate, 5 mg l⁻¹; cobalt (II) chloride hexahydrate, 1.7 mg l⁻¹; copper (II) sulphate penta-hydrate, 1.6 mg l⁻¹; sodium molybdate dehydrate, 1.5 mg l⁻¹; biotin, 1 mg l⁻¹; nicotinic acid, 1 mg l⁻¹; pyridoxine, 1 mg l⁻¹; riboflavin, 1 mg l⁻¹; thiamine, 1 mg l⁻¹; sodium nitrate, 6 g l⁻¹; potassium chloride, 0.5 g l⁻¹; magnesium sulfate, 0.5 g l⁻¹; monopotassium phosphate, 1.5 g l⁻¹; 20 g l⁻¹) agar plates and grown for 10−15 days at 25 °C. Subsequently spores were harvested by washing the plates with liquid CM. Cells were sedimented at 1000 g Heraeus Biofuge Stratos benchtop centrifuge (Kendro Laboratory Products) for 10 min and incubated in fresh CM for 2−3 days, 25 °C/100 rpm, before use in experiments.

### Identification of *Z. tritici* homologues and bioinformatics
To identify homologues of the chosen reporter proteins, we screened the published sequence of *Z. tritici* (http://genome.jgi.doe.gov/Mycgr3/ Mycgr3.home.html), using the provided BLASTP function and the *S. cerevisiae* protein sequences of Erg5 (NCBI accession number: NP_013728.1), Erg6 (NCBI accession number: NP_013706.1), Sod2

 

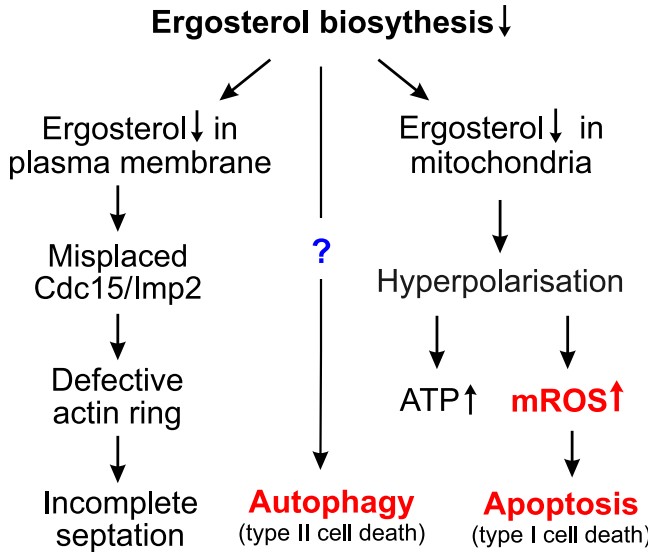

**Ergosterol biosynthesis↓**

**Fig. 9 | The MoA of ergosterol biosynthesis inhibitors.** Ergosterol is enriched in the plasma membrane and the inner mitochondrial membrane. Inhibition of ergosterol biosynthesis reduces ergosterol levels in the plasma membrane (PM), which impacts on the formation of actin rings, leading to incomplete septum formation. Inhibition of ergosterol biosynthesis in mitochondria results in hyperpolarization of the IMM, which raises ATP levels in the cytoplasm, but also increases mROS production at respiration complex I. mROS oxidises lipids and proteins in mitochondria and this is harmful itself. Moreover, increased mROS triggers 'suicidal' apoptosis in the pathogen cell (type I programmed cell death). In parallel, changes in the lipid composition of membranes at autophagosome formation sites may induces autophagy (type II programmed cell death). However, other stimuli, such as DNA damage or nutrient shortage due to altered transport over the plasma membrane may trigger autophagy ('?'). Simultaneous inhibition of mROS, apoptosis and autophagy 'neutralise' azoles, suggesting that the activation of both programmed cell death pathways, most likely in combination with oxidising mROS underpins the fungicidal activity of azoles in crop pathogens. Note that high azole concentrations induce PM depolarisation in vitro. Colour-blind friendly images are available in the Supplementary Fig. 20.

(NCBI accession number: AJU21583.1), Hyr1 (NCBI accession number: EGA61848.1), Mca1 (NCBI accession number: EDN63533.1), Rny1 (NCBI accession number: QHB12138.1), Atg1 (NCBI accession number: QHB08479.1), Atg8 (NCBI accession number: QHB06698.1). Sequences were obtained from the NCBI server (https://www.ncbi.nlm.nih.gov/) and comparison was done using EMBOSS Needle (http://www.ebi.ac.uk/Tools/psa/emboss_needle/) and domain structures were analysed in PfamScan (https://www.ebi.ac.uk/Tools/pfa/pfamscan/).

**Molecular cloning**

Vectors were generated by in vivo recombination in *S. cerevisiae* DS94 (MATα, *ura3-52*, *trp1−1*, *leu2-3*, *his3−111* and *lys2-801*). *S. cerevisiae* DS94 cells were grown in 3 ml YPD at 28 °C, 12 h at 200 rpm and used to inoculate 50 ml YPD. After an additional 5 h at 28 °C, 200 rpm, cells were harvested by centrifugation at $672 \times g$ for 5 min, washed with 5 ml sterile distilled water, and after re-centrifugation, were re-suspended in 300 µl sterile water. For in vivo recombination, ~100 ng in 4 µl volume of each purified DNA fragments using silica glass suspension were mixed with 50 µl salmon sperm DNA (2 mg ml⁻¹ stock; Sigma-Aldrich), 50 µl (~15 million) *S. cerevisiae* DS94 cells, 32 µl of 1 M lithium acetate and 240 µl of 50% (v v⁻¹) PEG 4000 (Sigma-Aldrich). Samples were gently mixed by pipetting up and down and incubated at 28 °C for 30 min. A heat shock was induced at 45 °C for 15 min, and samples were centrifuged at $400 \times g$ for 2 min at room temperature. Sedimented cells were re-suspended in 150 µl sterile distilled water and were plated onto yeast synthetic drop-out medium, which lacks uracil (yeast nitrogen base without amino acids and ammonium sulphate,

1.7 g l⁻¹; ammonium sulphate, 5 g l⁻¹; casein hydrolysate, 5 g l⁻¹; tryptophan, 20 mg l⁻¹; agar, 20 g l⁻¹), followed by 2 d incubation at 28 °C. Colony PCR was performed on yeast cells by using DreamTaq DNA polymerase (Thermo Scientific, Leicestershire, UK) in 20 µl total volume. Plasmid DNA was isolated from the positive yeast colonies and 10 µl of plasmid DNA isolated from the *S. cerevisiae* was transformed into and amplified in *E. coli* strain DH5α. Finally, the plasmid DNA was isolated from the *E. coli* colonies and further confirmed by restriction analysis. All restriction enzymes were obtained from New England Biolabs (Herts, UK). Each plasmid was generated as described below.

**pHΔAtg4.** Vector pHΔAtg4 is used to replace the endogenous *atg4* gene with the hygromycin resistance cassette. This vector was generated using 9760 bp fragment of pCGEN-YR (digested with *Xba*I and *Zra*I), 1000 bp of *atg4* promoter (amplified with primers SK-Sep-743 and SK-Sep-744 from IPO323 genomic DNA; see Supplementary Table 5 for all cloning primers), 1806 bp of hygromycin resistance cassette (amplified with primers SK136 and SK137 from plasmid pChygYR) and 1000 bp of the non-coding 3′ region of *atg4* (amplified with primers SK-Sep-745 and SK-Sep-746 from IPO323 genomic DNA).

**pHImp2ZtGFP.** This plasmid contains codon-optimised *ztgfp* fused to the *imp2* gene and is designed for random ectopic integration into the genome of *Z. tritici*. This vector was generated using 14947 bp fragment of pHHis1ZtGFP[24] (digested with *Pml*I), and 1000 bp *imp2* promoter and 2905 bp *imp2* full-length gene (amplified with primers SK-Sep-942 and SK-Sep-943 from IPO323 genomic DNA).

**pGHis1mCherrry.** This plasmid contains codon-optimised *ztgfp* fused to the full-length *zthis1* under the control of constitutive *zttub2* promoter and terminator sequences for random ectopic integration into the genome of *Z. tritici* by using geneticin (G418) as selection agent. This plasmid was generated using the 13,552 bp fragment of pGmCherry-Sso1[24] (digested with *Bsr*GI and *Xho*I), DNA fragment consists of 977 bp *zttub2* promoter and 1235 bp full-length *zthis1* gene (amplified with primers SK-Sep-46 and SK-Sep-949 from plasmid pCHis1ZtGFP[24], 720 bp mCherry (amplified with primers SK-Sep-215 and SK-Sep-90 from plasmid pGmCherry-Sso1[24]. The plasmid pGHis1mCherrry was random ectopically integrated into strain IPO323_eGAtg8[24] to give strain IPO323_eGAtg8_His1mCh.

**pCPerg5ZtGFP.** This plasmid contains codon-optimised *ztgfp* under the control of the promoter of a gene encoding sterol C-22 desaturase Erg5. The plasmid was generated using the 13,083 bp fragment of plasmid pCZtGFP (*Pml*I-digested) and 1500 bp of *erg5* promoter (amplified with primers SK-Sep-808 and SK-Sep-809 from IPO323 genomic DNA). The plasmid pCPerg5ZtGFP was integrated into the *sdi1* locus of strain IPO323_mChSso1[26] to give strain IPO323_mChSso1_Perg5ZtG.

**pCPerg6ZtGFP.** This plasmid contains codon-optimised *ztgfp* under the control of the promoter of a gene encoding sterol 24-C-methyltransferase Erg6. The plasmid was generated using the 13,083 bp fragment of plasmid pCZtGFP (*Pml*I-digested) and 1500 bp of *erg6* promoter (amplified with primers SK-Sep-810 and SK-Sep-811 from IPO323 genomic DNA). The plasmid pCPerg6ZtGFP was integrated into the *sdi1* locus of strain IPO323_mChSso1 to give strain IPO323_mChSso1_Perg6ZtG.

**pCPsod2ZtGFP.** This plasmid contains codon-optimised *ztgfp* under the control of the promoter of a gene encoding superoxide dismutase Sod2. The plasmid was generated using the 13,083 bp fragment of plasmid pCZtGFP (*Pml*I-digested) and 1500 bp of *sod2* promoter (amplified with primers SK-Sep-802 and SK-Sep-803 from IPO323 genomic DNA). The plasmid pCPsod2ZtGFP was integrated into the

*sdi1* locus of strain IPO323_mChSso1 to give strain IPO323_mChSso1_P*sod2*ZtG.

**pCP*hyr1*ZtGFP.** This plasmid contains codon-optimised *ztgfp* under the control of the promoter of a gene encoding glutathione peroxidase Hyr1. The plasmid was generated using the 13,083 bp fragment of plasmid pCZtGFP (*Pml*I-digested) and 1500 bp of *hyr1* promoter (amplified with primers SK-Sep-804 and SK-Sep-805 from IPO323 genomic DNA). The plasmid pCP*hyr1*ZtGFP was integrated into the *sdi1* locus of strain IPO323_mChSso1 to give strain IPO323_mChSso1_P*hyr1*ZtG.

**pCP*mca1*ZtGFP.** This plasmid contains codon-optimised *ztgfp* under the control of the promoter of a gene encoding metacaspase Mca1. The plasmid was generated using the 13,083 bp fragment of plasmid pCZtGFP (*Pml*I-digested) and 1500 bp of *mca1* promoter (amplified with primers SK-Sep-786 and SK-Sep-787 from IPO323 genomic DNA). The plasmid pCP*mca1*ZtGFP was integrated into the *sdi1* locus of strain IPO323_mChSso1 to give strain IPO323_mChSso1_P*mca1*ZtG.

**pCP*rny1*ZtGFP.** This plasmid contains codon-optimised *ztgfp* under the control of the promoter of a gene encoding RNase in yeast Rny1. The plasmid was generated using the 13,083 bp fragment of plasmid pCZtGFP (*Pml*I-digested) and 1500 bp of *rny1* promoter (amplified with primers SK-Sep-788 and SK-Sep-789 from IPO323 genomic DNA). The plasmid pCP*rny1*ZtGFP was integrated into the *sdi1* locus of strain IPO323_mChSso1 to give strain IPO323_mChSso1_P*rny1*ZtG.

**pCP*atg1*ZtGFP.** This plasmid contains codon-optimised *ztgfp* under the control of the promoter of a gene encoding serine-threonine protein kinase Atg1. The plasmid was generated using the 13,083 bp fragment of plasmid pCZtGFP (*Pml*I-digested) and 1500 bp of *atg1* promoter (amplified with primers SK-Sep-776 and SK-Sep-777 from IPO323 genomic DNA). The plasmid pCP*atg1*ZtGFP was integrated into the *sdi1* locus of strain IPO323_mChSso1 to give strain IPO323_mChSso1_P*atg1*ZtG.

**pCP*atg8*ZtGFP.** This plasmid contains codon-optimised *ztgfp* under the control of the promoter of a gene encoding ubiquitin-like protein Atg8. The plasmid was generated using the 13,083 bp fragment of plasmid pCZtGFP (*Pml*I-digested) and 1500 bp of *atg8* promoter (amplified with primers SK-Sep-780 and SK-Sep-781 from IPO323 genomic DNA). The plasmid pCP*atg8*ZtGFP was integrated into the *sdi1* locus of strain IPO323_mChSso1 to give strain IPO323_mChSso1_P*atg8*ZtG.

Plasmids descriptions and references are summarised in Supplementary Table 2.

### *Z. tritici* transformation and molecular analysis of transformants

Vectors were transformed into *A. tumefaciens* strain EHA105 and *A. tumefaciens*- mediated *Z. tritici* transformation performed as described[24]. 1 μl vector DNA was added to $1 \times 10^5$ *A. tumefaciens* competent cells, heat shocked at 37 °C for 5 min and retained on ice for 5 min. After the addition of 1 ml DYT medium, tubes were incubated at 28 °C, at 200 rpm for 3–4 h, followed by plating on DYT agar medium, supplemented with 20 μg ml$^{-1}$ rifampicin (Melford, Ipswich, UK) and 50 μg ml$^{-1}$ kanamycin (Sigma-Aldrich). Positive transformants were identified by colony PCR and grown in 10 ml DYT medium/20 μg ml$^{-1}$ rifampicin/ 50 μg ml$^{-1}$ kanamycin overnight at 28 °C, 200 rpm. Cells from this culture were diluted to an optical density at 600 nm of 0.15 in 10 ml *Agrobacterium* induction medium (AIM; 10 mM KH$_2$PO$_4$, 10 mM K$_2$HPO$_4$, 2.5 mM NaCl, 2 mM MgSO$_4$·7H$_2$O, 0.7 mM CaCl$_2$, 9 mM FeSO$_4$, 4 mM (NH$_4$)$_2$SO$_4$, 10 mM glucose, 0.5% (w v$^{-1}$) glycerol, 40 mM MES buffer, 1 l H$_2$O, pH 5.6), supplemented with 200 μM acetosyringone (Sigma-Aldrich) and grown at 28 °C, 200 rpm, until the optical density reached to 0.3–0.35 (-3 h). *Z. tritici* spores were harvested from 5-day old YPD plates and diluted to a concentration of $1 \times 10^7$ ml$^{-1}$ in AIM. Equal volumes of the *Z. tritici* and *A. tumefaciens* cultures were

combined and 200 μl of this mixture was plated onto nitrocellulose filters (AA packaging limited, Preston, UK), overlaid onto AIM agar plates (AIM, 2% agar, w v$^{-1}$), supplemented with 200 μM acetosyringone. After growth for 3 days at 18 °C, the filters were transferred onto Czapek Dox agar plates (Oxoid, Basingstoke, UK), containing 100 μg ml$^{-1}$ cefotaxime (Melford), 100 μg ml$^{-1}$ timentin (Melford) and 200 μg ml$^{-1}$ hygromycin (Invitrogen, Toulouse, France) and incubated at 18° for 8–12 days. Colonies were transferred onto YPD agar plates, containing 100 μg ml$^{-1}$ cefotaxime, 100 μg ml$^{-1}$ timentin and 200 μg ml$^{-1}$ hygromycin and grown at 18 °C for 3–4 days.

The vector pHΔAtg4 was integrated into the *atg4* locus of IPO323_eGAtg8 and IPO323_eGSso1[24] resulting in strains IPO323_ΔAtg4_eGAtg8 and IPO323_ΔAtg4_eGSso1. To co-visualise the ZtImp2 and PM, vector pGmCherrySso1[24] was transformed into strain IPO323 resulting in IPO323_mChSso1. Subsequently, vector pHImp2ZtGFP was transformed into strain IPO323_mChSso1 resulting in strain IPO323_mChSso1_Imp2ZtG. To co-visualise autophagosomes and nucleus, vector pGHis1mCherry (this study) was transformed into strain IPO323_eGAtg8[24] to give strain IPO323_eGAtg8_His1mCh. Vectors pCP*erg5*ZtGFP, pCP*erg6*ZtGFP, pCP*sod2*ZtGFP, pCP*hyr1*ZtGFP, pCP*mca1*ZtGFP, pCP*rny1*ZtGFP, pCP*atg1*ZtGFP and pCP*atg8*ZtGFP were integrated into the *sdi1* locus of strain IPO323_mChSso1[26] to give strains IPO323_mChSso1_P*erg5*ZtG, IPO323_mChSso1_P*erg6*ZtG, IPO323_mChSso1_P*sod2*ZtG, IPO323_mChSso1_P*hyr1*ZtG, IPO323_mChSso1_P*mca1*ZtG, IPO323_mChSso1_P*rny1*ZtG, IPO323_mChSso1_P*atg1*ZtG and IPO323_mChSso1_P*atg8*ZtG respectively.

Integration of vector pHΔAtg4 into the native locus of *atg4* was confirmed by Southern blot. To this end, *Z. tritici* transformants obtained with pHΔAtg4 and wildtype IPO323 were grown in YG medium for 3 days at 18 °C with 200 rpm. Approximately 3 μg of genomic DNA was used for restriction digestion (*Xho*I for *atg4* locus) and separated on a 1.0% agarose gel and capillary transferred to a Hybond-N membrane (GE Healthcare, Little Chalfont, United Kingdom). 1000 bp *atg4* probe (amplified with primers SK-Sep-743 and SK-Sep-744) was generated by using PCR DIG Probe Synthesis kit (Sigma-Aldrich, UK). Hybridisations were performed at 62 °C for overnight and autoradiographs were developed after an appropriate period.

### Plate colony growth essays

Sensitivity of *Z. tritici* IPO323 to azoles was determined as described previously[71]. In brief, YPD agar plates were supplemented with various azole concentrations. The number of *Z. tritici* IPO323 spores in 5 day-old cultures was determined using a Cellometer Auto 1000 cell counter (Nexcelom Biosciences, Lawrence, USA). A dilution series was prepared ($10^6$ cells ml$^{-1}$, $0.5 \times 10^6$ ml$^{-1}$, $0.25 \times 10^6$ ml$^{-1}$, $1.25 \times 10^5$ ml$^{-1}$ and $6.25 \times 10^4$ ml$^{-1}$) and 5 μl of this cell suspension were placed on the plates, followed by incubation for 5 days at 18 °C. Digital images of plates were obtained using an Epson Perfection V750 Pro scanner (Epson, Hemel Hempstead, UK). Growth inhibition was assessed by measuring the integrated intensity of the third highest dilution. Digital images were converted to grey-scale using Photoshop CS6 (Adobe Inc., San Jose, USA), an area of interest was drawn around the colony and the integrated intensity of all pixels in this area determined. Next, the area of interest was moved to a fungal cell-free region and the integrated intensity of this background was measured. After correcting for this background, the values for colony formation on control plates, were set to 100% and all other measurements were compared. The resulting values represented the relative cell density (=colony brightness) and were plotted and analysed in GraphPad Prism 6 to obtain fungicide dose–response curve.

The sensitivity of *M. oryzae* Guy11 to azoles was determined using a mycelial assay. Potato Dextrose Agar (PDA; potato dextrose broth, 24 g ml$^{-1}$, agar, 20 g ml$^{-1}$) plates were supplemented with a series of concentration of azoles. Mycelial plugs of 5 mm diameter were excised from the growing edges of 5 day-old *M. oryzae* Guy11 colonies and

transferred to the centre of PDA plates. Colony diameters were measured perpendicularly after 3 days of incubation at 25 °C. The resulting values were plotted and analysed in GraphPad Prism 6 to obtain fungicide dose-response curves.

### Electron microscopy

Liquid-grown cells grown as liquid cultures were fixed in suspension in 2% (w v$^{-1}$) glutaraldehyde and 2% (w v$^{-1}$) paraformaldehyde in either 0.1 M sodium cacodylate or 0.1 M PIPES buffer (pH 7.2) for 2 h at room temperature then washed 3 × 5 min in the same buffer before post-fixation in 2% (w v$^{-1}$) potassium permanganate in deionised water for 1 h on a rotator. After 3 × 5 washes in deionised water the cells were dehydrated in a graded ethanol series (30%, 50%, 70%, 80%, 90%, 95% ethanol, 10 min per step then 2 × 20 min in 100% ethanol) and subsequently embedded in SPURR resin. 60 nm ultrathin sections were collected on piliform-coated 100 mesh copper EM grids (Agar Scientific, Stansted, UK), contrasted in Reynold's lead citrate for 10 min before inspection using a JEOL JEM 1400 transmission electron microscope operated at 120 kV. Images were taken with a digital camera (ES1000W, Gatan, Abingdon, UK).

### Fluorescence microscopy

Cells were observed using a motorised inverted microscope (IX81, IX83; Olympus, Hamburg, Germany), equipped with a PlanApo 100×/1.45 Oil TIRF or UPlanSApo 60×/1.35 Oil objective (Olympus). Fluorescent proteins or fluorescent dyes were excited using a VS-LMS4 Laser Merge System with 75 mW solid-state lasers (488 nm, 561 nm; Visitron Systems, Puchheim, Germany). For photo-bleaching and laser-induced rupture experiments, a 405 nm/60 mW diode laser was used. This was coupled into the light path by an OSI-IX 71 adaptor (Visitron System) and controlled by a UGA-40 unit and a VisiFRAP 2D FRAP control software (Visitron System). Z-stacks were generated by using an objective Piezo (Piezosystem Jena GmbH, Jena, Germany). Simultaneous observation of red and green fluorescence was performed using a dual-imager (Dual-View 2 Multichannel Imaging System; Photometrics, Tucson, USA) equipped with a dual-line beam splitter (z491/561; Chroma Technology Corp., Bellows Falls, USA), with an emission beam splitter (565 DCXR; Chroma Technology Corp.), an ET-Band pass 525/50 (Chroma Technology Corp.) and a single band pass filter (BrightLine HC 617/73; Semrock, New York, USA). Images were captured using a CoolSNAP HQ2 or a PRIME-BSI-Express camera (Photometrics). All parts of the system were under the control of the software package VisiView (Visitron System). MetaMorph 7.8.x (Molecular Devices, Wokingham, UK) was used for all image processing.

Analysis of fungicidal activity of epoxiconazole on wheat leaves used a TCS SP8 confocal laser scanning microscope (Leica, Wetzlar, Germany), equipped with various lasers (argon, blue diode, DPSS561, HeNe594 and a HeNe633), two conventional PMT detectors and three hybrid detectors. An HC PL APO 63×/1.2 water objective was used for all imaging. Image processing used Leica LAS AF X 3.5.2.18963 software.

To quantify the expression of the reporter constructs *in planta*, a spinning disc confocal microscopy consisting of a VisiScope Confocal Cell Explorer (Visitron System) an IX81 motorised inverted microscope (Olympus), a CSU-X1 Spinning Disc unit (Yokogawa, Japan), a UPlan-SApo63/1.35 oil objective (Olympus). Fluorescent tags were excited using a VSLMS6 Laser Merge System (488 nm/100 mW and 561 nm/100 mW, Visitron System). Images were taken using an OptoSplit II LS Image Splitter (Cairn Research Limited, UK) and a Photometrics CoolSNAP HQ2 camera (Photometrics). The system was controlled by the software package VisiView (Visitron System).

### Fungicides and inhibitors used in this study

Epoxiconazole and metconazole were used as unformulated and pure compounds. These fungicides, ergosterol and all other inhibitors, used in this study (rotenone, 3-MA, antimycin A) were purchased from

Sigma-Aldrich (Poole, UK). All experiments involving epoxiconazole and metconazole in *Z. tritici* used stock solutions at 0.01–1 mg ml$^{-1}$ azole content, dissolved in methanol. A final concentration of 0.01 μg ml$^{-1}$ or 1 μg ml$^{-1}$ was usually incubated for 24 h or as described in the figure legends. The mitochondrial respiration complex inhibitors rotenone and antimycin A as well as the autophagy inhibitor methyladenine (3-MA) were dissolved in dimethylsulfoxide (Sigma-Aldrich) at stock concentrations of 100 mM, 10 mM or 1 mM, respectively, were used. Final concentrations of rotenone, antimycin A and 3-methyladenine (3-MA) were 100, 10 and 5 μM, respectively. All experiments involving terbinafine, tolfonate, fluconazole and fenpropimorph used stock methanol-based solutions at concentrations of 5 mg ml$^{-1}$, 50 mg ml$^{-1}$, 15 mg ml$^{-1}$ and 5 mg ml$^{-1}$. For all experiments, a final concentration of 1/1000 of the stocks was used. To test the effect of external ergosterol in epoxiconazole-treated *Z. tritici* cells 5 μg ml$^{-1}$ exogenous ergosterol was added. Experiments using *M. oryzae* were performed using epoxiconazole and metconazole containing methanol-based stock solutions of 1–25 mg ml$^{-1}$ and a final concentration of 10 μg ml$^{-1}$. All control experiments contained an equivalent volume of the solvent (methanol or dimethylsulfoxide).

### Plant-growth conditions and fungicide spraying

*Triticum aestivum* cultivar Galaxy, was grown in a Fitotron SGC 120 growth chamber (Weiss Technik UK, Loughborough, UK), using 14 h light (intensity 500 μmol of PAR), 24 °C, 80% RH (relative humidity); 10 h dark, 20 °C, 60% RH with automated watering for 14 days. To test the ability of azoles to kill *Z. tritici*, the lower surface of the second leaf of 14-day-old wheat plants was sprayed with 0.1 μg ml$^{-1}$ epoxiconazole (stocks 0.1 mg ml$^{-1}$ in methanol; 1 ml for 10 leaves) and 0.04% (v v$^{-1}$) Tween 20 (Sigma-Aldrich), using a Voilamart AS18 airbrush/compressor (Hydirect, Guang Dong, China), set at 5 psi. The control plants were sprayed with the equivalent volumes of the solvent methanol. After 24 h growth, plants were sprayed at 5 psi with 1 ml of cell suspension of *Z. tritici* 5 × 10$^5$ cells per ml. Pathogen cell numbers were determined using a Cellometer Auto1000 cell counter (Nexcelom Biosciences, Lawrence, USA). Inoculated plants were grown for additional 2 – 5 days, followed by staining with DiBAC$_4$(3) or propidium iodide (see below) and imaging using a laser scanning confocal or spinning disc microscope (see above).

### Test for fungistatic effect of azoles

IPO323_eGSso1 conidia were grown in YG liquid medium for 2 days at 18 °C, 200 rpm and adjusted to ~0.7 × 10$^6$ spores ml$^{-1}$ using Cellometer Auto 1000 cell counter; Nexcelom Biosciences. The number of cells per spore was determined by observing eGFP-Sso1-labelled PM using fluorescence microscopy. 5 ml of this pre-culture were treated with 0.01 or 1 μg ml$^{-1}$ azole for 2–8 h and spore and cell numbers per counted as described.

### Test for a fungicidal effect of azoles and mortality essays

(1) in vitro: A 48 h-old pre-culture of *Z. tritici* strain IPO323_eGSso1 was adjusted to an OD$_{600}$ of 0.4 and incubated for 1–4 days with various concentrations of epoxiconazole or metconazole at 18 °C with 200 rpm. In all cases, cells treated with 0.1% (v v$^{-1}$) methanol were used as control. Spores of *M. oryzae* wildtype strain Guy11 were grown for 48 h in CM. This pre-culture was divided into 5 ml aliquots and incubated for 1 or 4 days with various concentrations of epoxiconazole or metconazole at 25 °C with 100 rpm. Dead cells were identified by adding 0.2 μl LIVE/DEAD™ Fixable Red Dead Cell Stain (L34972; Thermo Fisher Scientific, Loughborough, UK) to 100 μl of cell suspension, followed by immediate microscopic investigation. To investigate the effect of azole-incubation time, cells of strain IPO323_eGSso1 were incubated for 24 h or 48 h with 1 μg ml$^{-1}$ epoxiconazole or metconazole at 18 °C, 200 rpm. Subsequently, cells were sedimented by centrifugation, washed three times with fresh YG medium and grown

for 2 or 3 days in the absence of fungicides. Cell mortality was defined as the proportion of dead cells in a cell population, treated with azoles or inhibitors. Individual cells in a spore of *Z. tritici* were identified by observation of eGFP-ZtSso1. In case of *M. oryzae* (strain Guy11), the tip cells of hyphae were analysed. Dead cells that contained red-fluorescent LIVE/DEAD™ dye were counted and their proportion amongst all tip cells was determined.

To assess the role of mROS, apoptosis and/or autophagy on epoxiconazole-induced cell mortality, IPO323_eGSso1 cells were treated with epoxiconazole, rotenone (Sigma-Aldrich), 3-MA (3-Methyladenine, Sigma-Aldrich) alone or in various combinations. Fresh YG medium, supplemented with the respective inhibitors and/or fungicides, was added after 3 days of growth. Dead cells were counted daily by staining 100 µl of the cell cultures with 0.2 µl propidium iodide (Sigma Aldrich; stock: 1 mg ml⁻¹ in ddH₂O; note that the amine-binding LIVE/DEAD™ dye interfered with rotenone).

(2) *In planta*: Plants were grown, treated with 0.1 µg ml⁻¹ epoxiconazole and inoculated with *Z. tritici* spores as described above. Leaf sections were placed on Carolina Observation Gel (Carolina Biological Supply Company, USA), followed by the addition of a drop of staining solution (1 µl propidium iodide stock in 1 ml ddH₂O). Leaf samples were imaged using laser scanning confocal microscopy.

## Visualisation of organelles and septa

To visualise nuclei, actin cytoskeleton and the septa in IPO323. *Z. tritici* cells expressing His1-ZtGFP, Lifeact-ZtGFP or eGFP-ZtSso1 were grown for 48 h in YG medium. The precultures were adjusted to an $OD_{600}$ of 0.4 and treated with the different chemistries for 24 h at 18 °C, 200 rpm. This was followed by acquisition of Z axis image stacks with a Z resolution of 0.2 µm and an exposure time of 150 ms. The 488 nm laser was used at 5–50% output power. The final images are maximum projections generated in MetaMorph (Molecular Devices). To visualise autophagosomes, liquid-grown cells were treated with epoxiconazole or metconazole for 1–5 days. Z-axis image stacks were acquired and the number of autophagosomes within a defined 3D space was analysed.

## Membrane fluidity

The effect of the azoles on the PM fluidity was analysed in strain IPO323_eGSso1. Prior to bleaching, a reference image of the middle plane of the cell was taken. Subsequently, ~2.5 µm of the GFP-labelled membrane was photo-bleached, using a 405 nm laser pulse of 50 ms at 10% intensity. Recovery was recorded immediately after at 30 s intervals. To measure fluorescent recovery, the average intensity in the bleached area was corrected for the adjacent background and compared to a background-corrected measurement of the GFP-sso1 intensity in unbleached regions of the PM. The average intensity over time in these regions was measured and the percentage intensity of the bleached region relative to the unbleached region was calculated.

## Plasma membrane depolarisation assay

(1) In vitro: To test the effect of epoxiconazole and metconazole on PM potential as an indication of PM integrity, 1 ml suspension of azole-treated cells of strain IPO323_mChSso1 was stained with 1 µl of the voltage-sensitive fluorescent dye DiBAC₄(3) (bis-(1,3-dibutylbarbituric acid) trimethine oxonol, Thermo Fisher Scientific, Loughborough, UK; stock: 20 mg ml⁻¹ in dimethylsulfoxide). Dead cells were identified by co-staining with 0.5 µl propidium iodide (Sigma Aldrich; stock: 1 mg ml⁻¹ in ddH₂O). After 5 min incubation at room temperature, cells were sedimented, washed with fresh YG medium, and imaged using fluorescence microscopy (see above) using the 488 nm laser at 5% and the 561 nm laser at 50% intensity. Cells were identified by observation of mCherry-ZtSso1. Dead cells that contained red-fluorescent propidium iodide and green-fluorescent DiBAC₄(3) were excluded from the analysis. The proportion of DiBAC₄(3)-positive cells amongst all living cells was determined.

(2) *In planta*: Plants were grown, treated with 0.1 or 1 µg ml⁻¹ epoxiconazole and inoculated with *Z. tritici* spores as described above. Leaf sections were placed on Carolina Observation Gel (Carolina Biological Supply Company, USA), followed by addition of a drop of staining solution (1 µl DiBAC₄(3) stock, 1 µl propidium iodide in 1 ml ddH₂O). Leaf samples were observed using spinning disc confocal microscopy and analysed as described above.

## Laser-induced cell rupture

*Z. tritici* strain IPO323_His1ZtG_mChSso1 was grown as described above. After 48 h, the $OD_{600}$ was adjusted at 0.4 and spores were incubated for 24 h with 0.01 µg ml⁻¹ epoxiconazole or metconazole. Laser-induced wounding was performed at the IX-81 fluorescent microscope, using a 200 ms light pulse of the 405 nm/60 mW diode laser at 100% output power. Image sequences of red and green fluorescence were acquired at 50 ms intervals as described above.

## Mitochondrial membrane potential

The changes in mitochondrial membrane potential were determined using tetramethylrhodamine, methyl ester, perchlorate (Image-iT™ TMRM Reagent, Thermo Fisher Scientific). To this end cultures of *Z. tritici* strain IPO323_eGSso1 were grown and treated with the azoles as described above. 1 µl of TMRM was added to 1 ml cell suspension, followed by 10 min incubation on aa SB2 Rotator (Bibby Scientific Limited, Stone, UK). TMRM fluorescence was imaged using the 561 nm laser at 20% intensity and the cell outline was detected by recoding GFP-ZtSso1 with the 488 nm laser. Subsequently, the average cellular intensity of TMRM fluorescence was measured and values were corrected for the cytoplasmic background. In the case of *M. oryzae* the mitochondrial membrane potential was analysed in tip cells of liquid-grown hyphae, using the 561 nm laser at 20% intensity and an exposure time of 150 ms. The average intensity of the first 10 µm of the hyphal tip cell was measured and the intensity values were corrected by the cytoplasmic fluorescent background.

## ATP detection in cytoplasmic extracts

To analyse the effect of azoles on cellular ATP levels, cells of *Z. tritici* IPO323 were grown in YG liquid medium for 48 h, 200 rpm. 20 ml of this culture were incubated for 24 h with 0.01 µg ml⁻¹ azoles. Corresponding amounts of the solvent methanol were used in control experiments. Subsequently, 4 ml cell suspension was centrifuged for 5 min at 2.4 g, using a Micro Star 17 R cooled centrifuge (VWR, Lutterworth, UK). Sedimented cells were resuspended in 100 µl of 10 mM Tris-HCl buffer (pH 7.0) and ruptured used acid-washed glass beads (425–600 µm bead size; Sigma- Aldrich) and an IKA Vibrax shaker (IKA, Staufen, Germany). The sample was centrifuged for 10 min at 12.1 g at 4 °C and the protein content in the supernatant was determined using a Qubit™ Protein Assay Kit (Q33212, Thermo Fisher Scientific) and a Qubit® 2.0 Fluorometer (Thermo Fisher Scientific). After samples were adjusted to the same protein concentration using 10 mM Tris-HCl buffer, ATP levels were detected in 5 µl of this supernatant, using the luciferase-based ATP Determination Kit (A22066; Thermo Fisher Scientific) as instructed by the supplier. Luminescence was detected using a GloMax® Discover plate reader (Promega, Wisconsin, USA).

## Production of reactive oxygen species detection

mROS was detected by staining cells of *Z. tritici* (strain IPO323_mChSso1) and *M. oryzae* (strain Guy11) with dihydrorhodamine 123 (DHR123, Sigma-Aldrich). Following incubation with the azoles, 1 µl DHR123 was added to 1 ml cell suspension and incubated for 15 minutes, at room temperature. Stained cells were placed onto a 2% (w v⁻¹) agar cushion and imaged using the described microscopic setup and the 488 nm laser at 10% output power (to detect DHR123 fluorescence) and 561 nm laser at 50% output power to detect mCherry-Sso1-labelled PM as an indication of cell boundaries. The average intensity of DHR123

fluorescence within the entire *Z. tritici* cells was measured in middle focal plane and corrected by the cytoplasmic background. In the case of *M. oryzae*, the average intensity of the apical 10 µm of hyphae was measured and corrected by the cytoplasmic background.

## Caspase activity

Metacaspase activity was visualised in treated cells of *Z. tritici* (strain IPO323_mChSso1) and *M. oryzae* (strain Guy11) using the CaspACE FITC-VAD-fmk In Situ Marker assay (Promega, Madison, USA). To this end, cells were grown and treated with azoles or solvent as described above. 100 µl cell suspension was incubated with 0.1 µl of FITC-VAD-fmk and 0.1 µl propidium iodide (Sigma Aldrich; stock: $1 mg ml^{-1}$ in $ddH_2O$) for 10 min at room temperature in the dark on a SB2 Rotator (Bibby Scientific Limited). Cells were harvested by centrifugation, using a Micro Star 17R cooled centrifuge (VWR, Lutterworth, UK) at 2.4 g for 5 min (*Z. tritici*) or 1.2 g for 5 min (*M. oryzae*). Cell sediments were re-suspended in 100 µl in YG or PDB, respectively. Cells were imaged with the 488 nm and 561 nm lasers at 40% and 80% laser intensity and an exposure time of 150 ms. Apoptotic cells were identified by green staining with the CaspACE FITC-VAD-fmk dye; propidium iodide-positive cells were excluded from the analysis.

## Annexin V staining

Annexin-V-FLUOS staining Kit (Roche, Basel, Switzerland) was used to visualise cell surface-exposed phosphatidylserine, following the manufacturer´s protocol. In brief, 1 ml of azole-treated *Z. tritici* cell suspension (strain IPO323_mChSso1) was centrifuged at 3.5 g for 5 min, using a Micro Star 17 R cooled centrifuge (VWR). The sedimented cells were re-suspended in 100 µl of 2% Annexin-V-FLUOS and 2% propidium iodide and incubated for 10 min at room temperature in the dark. Cells were washed twice with YG medium and imaged using the 488 nm laser at 40% and the 561 nm laser at 80% output power. Apoptotic cells were identified by their green-fluorescence; propidium iodide-positive cells were excluded from the analysis.

## *M. oryzae* spore germination assays

*M. oryzae* strain Guy 11 was grown on CM agar plates (see above) at 25 °C for 10–15 days, spores were harvested by washing the plates with sterile water, followed by filtration through two layers of sterile Miracloth (Calbiochem, Nottingham, UK) and centrifugation at 3.5 g for 5 min. The pellet was re-suspended in sterile water and placed onto hydrophobic glass coverslips (Menzel- Glässer, Thermo Fisher Scientific, Loughborough, UK) and incubated in a wet chamber for 30 min at 25 °C. The spores containing glass coverslips were flowed upside down on sterile water containing $10 µg ml^{-1}$ azole or the equal amount of the solvent methanol for 4–5 h. Glass coverslips were mounted on microscope slides using Vaseline and imaged using an IX81 microscope.

## Statistical analysis and data presentation

All measurements were done in raw 14-bit images, using MetaMorph 7.8 (Molecular Devices). Data calculation was performed in Excel (Microsoft, Redmond, USA) or Prism9 (GraphPad Software, San Diego, USA). All statistical testing was performed in Prism9. Data sets with a sample size of ≤6 were assumed to be normal distributed and are shown as mean ± standard error of the mean. Their statistical comparison used unpaired two-tailed Student's *t* testing with Welch's correction or one-way ANOVA testing. Data sets with a sample size of $n > 6$ were tested for normal distribution using Shapiro-Wilk testing. In case at least one data set did not pass the normality test ($P < 0.5$), the data were presented as Whiskers' plots, with 25th/75th percentiles indicated as blue lines, median as a red line, and minimum and maximum values as whiskers ends. These data sets were tested using non-parametric two-tailed Mann-Whitney test. All test results are included in the figure legend. Full statistical information, including median,

25th/75th percentile, minimum/maximum values are provided in the Source Data file. All graphs were generated in Prism9 and modified in CorelDraw X6 (Corel Corporation, Ottawa, Canada). Acquired images were adjusted in brightness and contrast and gamma, using MetaMorph. Colour-blind friendly images were generated using Visolve Deflector 4.5.1 (Ryobi Systems Co., Ltd., Japan; https://www.ryobi.co.jp/products/visolve/en/).

## Reporting summary

Further information on research design is available in the Nature Portfolio Reporting Summary linked to this article.

## Data availability

The authors confirm that all relevant data are included in the paper or in the Supplementary Information file. The source data underlying Fig. 1a–i (Supp. Fig. 1a-i), 2a–j (Supp. Fig. 4a-j), 3a–e (Supp. Fig. 6a-e), 4a–j (Supp. Fig. 8a-j), 5a–n (Supp. Fig. 10a-n), 6a–g (Supp. Fig. 13a-g), 7a-n (Supp. Fig. 14a-n), 8 b-f (Supp. Fig. 10b-f) and Supplementary Figs. 2a-d, 3a-c, 5a, b, 7a-c, 9a-e, 11a-i, 12a-d, 15, 16 and 19a-e are provided as a Source Data file. Additional information is available from the authors upon request. Source data are provided with this paper.

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

## Acknowledgements

We thank Dr. David Vela for acquiring data and Dr. Christian Hacker, Bioimaging Centre at the University of Exeter, for electron microscopy support. This work was funded, in part, by BBSRC grants BB/I025956/1 and BB/P018335/1 awarded to principal investigator G.S. For the purpose of open access, the author has applied a 'Creative Commons Attribution (CC BY) licence to any Author Accepted Manuscript version arising from this submission.

## Author contributions

G.S. conceived and coordinated the project, wrote the manuscript, prepared all figures and analysed data. S.K. performed all molecular cloning and *Z. tritici* strain generation. M.S. performed confocal and epi-fluorescence microscopy, analysed data. All authors contributed to the writing of the Methods section.

## Competing interests

The authors declare no competing interests.
