## [Peer Review File · Nature Communications]

Azoles activate type I and type II programmed cell death pathways in crop pathogenic fungiReviewer #1 (Remarks to the Author):

This manuscript by Schuster et al., describes a novel mechanism of Azoles to kill pathogenic fungi (from fungistatic to fungitoxic), specifically in this case *Z. tritici* that causes Septoria tritici blotch. This fungitoxic mechanism includes inducing incomplete septation, apoptosis and autophagy. Overall the paper is very well written and the figures are well crafted/presented and are convincing. I consider the referencing to be fine. I do not have too many additional comments on the paper. However, the authors should consider the following.

Major comments:

-Is epoxiconazole or metconazole itself degraded into other small molecule metabolites within the fungal cells? The reason for cell death maybe caused by these small molecule metabolites, not the original drug itself. It's important to note that I'm referring to whether the drug itself is metabolized, and not whether the drug causes changes in the fungus' metabolites (such as the mentioned methyl-sterols in the article). This question is critical because the author does not know whether it is the two compounds themselves or their metabolites that lead to the cell death phenotype.

- epoxiconazole and metconazole are fungicides used to control fungi in plants. Therefore, it is important to consider whether long-term use of these chemicals can lead to plant toxicity and death. Additionally, it is worth investigating if the fungi isolated from treated plants display the same phenotypes under laboratory conditions. I suggest conducting experiments where the plants are treated with the chemicals for 1-2 days, then inoculating them with the fungi and observing any related phenotypic changes.

-The author did not observe the growth status of the mycelium (infectious hyphae) when the pathogen interacts with the host. I suggest inoculating the mycelium after applying the chemicals to the plant for 1-2 days, and observing the growth status of the infectious hyphae. Currently, the author is more focused on the vegetable hyphae, which may not be consistent with the actual situation.

Minor comments:

-How is cell death determined in Figure 1a? The control in Figure 1d should also be stained with LIVE/DEAD.

-Is the drug Epoxiconazole or Metconazole used as a raw material or as a formulated product? The text needs to indicate whether it is the active ingredient used.

-Rice blast fungus infects host plants mainly relies on conidia germination and appressorium formation. Compared to the phenotype of vegetable growth, the author should pay more attention to the phenotypes of these two stages.

Reviewer #2 (Remarks to the Author):

In current study, Schuster et al. elucidate the mode of action of epoxiconazole and metconazole in the wheat pathogen *Zyloseptoria tritici* and the rice blast fungus *Magnaporthe oryzae*. The authors show that both azoles have fungicidal activity and reduce fluidity, but not integrity, of the plasma membrane. This impairs localization of Cdc15-like F-BAR proteins, resulting in defective actin ring assembly and incomplete septation. However, genetic and pharmacological experiments show that azole lethality may be due to a combination of reactive oxygen species-induced apoptosis and autophagy. Simultaneous inhibition of both programmed cell death pathways abolishes azole-induced cell death. Thus, the authors make a conclusion that ergosterol synthesis-targeting fungicides kill crop pathogens by activation of two cell death pathways. Here are my comments:

1) Line87-88: Why did the author use 1 ug/ml azole, not other concentrations? If you use a lower concentration (e.g. 0.01, that was used in Figs. 2-5), what will happen? Is the fungistatic or fungicidal activity related to azole concentration?

2) Fig. 1: Since *Z. tritici* is a filamentous fungus, how to account cell number in this experiment?

3) Line 260-261, and Fig. 5a: What are the eGFP-Atg8-positive autophagocytic organelles? To verify the conclusion that azoles induce autophagy, the authors have to conduct a GFP-Atg8 cleavage analyses with Northern blotting which is a classic method for autophagy research. Importantly, the autophagic process is dynamic, only microscopic observation can not display the global autophagic status in each sample.

4) Lines 295-298: The authors found that Δ atg4 mutants were strongly reduced in azole-induced mortality. Would induction of autophagy (such as nitrogen starvation treatment) increase azole-induced mortality?

5) Line 312-313: The Δ atg4 mutants still formed a low number of phagophores, and 25 μ M 3-MA was unable to block autophagy completely, the author should test sensitivity of Δ atg8 to azole since Agt8 is essential for autophagy in fungi.

6) Fig 8. Is *Z. tritici* able to take up exogenous ergosterol? (*S. cerevisiae* cells can take up exogenous sterols under anaerobic conditions). The logic of Fig. 8 shows that septation and cell death are depended on ergosterol biosynthesis. Is this true absolutely? If *Z. tritici* can take up exogenous ergosterol, when *Z. tritici* is supplemented with exogenous ergosterol, can azoles still induce cell death and incomplete septation in the fungi?

Reviewer #3 (Remarks to the Author):

What are the noteworthy results?

In their manuscript, Martin Schuster et al. describe the impact of azole antifungals on the cell biology of the plant pathogens *Zymoseptoria tritici* and *Magnaporthe oryzae*. Using a series of elegant microscopy-based experiments, they show that azoles are 1) fungicidal against *Z. tritici*, 2) induce apoptosis in both pathogens, 3) induce autophagy in both pathogens. Finally, they show that other sterol biosynthesis inhibitors (i.e., allylamines, thiocarbamate and morpholines) cause similar effects.

Will the work be of significance to the field and related fields? How does it compare to the established literature? If the work is not original, please provide relevant references.

I think this work adds significantly to the understanding of the mode of action of azole antifungals. It has been reported previously that azoles trigger ROS production which may even be associated with apoptosis. However, none of the previous studies explored this in such detail as it was done in the present study, and the results remained too general and superficial.

Related work:

Ferreira GF, Baltazar Lde M, Santos JR, Monteiro AS, Fraga LA, Resende-Stoianoff MA, Santos DA. The role of oxidative and nitrosative bursts caused by azoles and amphotericin B against the fungal pathogen *Cryptococcus gattii*. *J Antimicrob Chemother.* 2013 Aug;68(8):1801-11.

Shekhova E, Kniemeyer O, Brakhage AA. Induction of Mitochondrial Reactive Oxygen Species Production by Itraconazole, Terbinafine, and Amphotericin B as a Mode of Action against *Aspergillus fumigatus*. *Antimicrob Agents Chemother.* 2017 Oct 24;61(11):e00978-17.

Lee W, Lee DG. Reactive oxygen species modulate itraconazole-induced apoptosis via mitochondrial disruption in *Candida albicans*. *Free Radic Res.* 2018 Jan;52(1):39-50.

Muñoz-Megías ML, Sánchez-Fresneda R, Solano F, Maicas S, Martínez-Esparza M, Argüelles JC. The antifungal effect induced by itraconazole in *Candida parapsilosis* largely depends on the oxidative stress generated at the mitochondria. *Curr Genet.* 2023 Jun;69(2-3):165-173.

Does the work support the conclusions and claims, or is additional evidence needed?

Overall, the experimental results support the conclusions very well. In my opinion, there is no additional evidence needed to support the conclusions per se. However, there are a few points that should be addressed with some simple experiments which could help to better interpret the results (see below).

Are there any flaws in the data analysis, interpretation and conclusions? Do these prohibit publication or require revision?

Major points:

1. Many experiments were performed at very low azole concentrations (0.01 µg per ml) which are presumably (?) below the minimal inhibitory concentrations (MICs) of the drugs. (Line 112-114: "Previous work has shown that the primary physiological effect of fungicides can be determined at low concentrations, where mortality is at ~10-20% and the majority of the cells are still alive") While the experimental results obtained under these conditions (apoptosis/autophagy) appear sound and are interesting, it raises the question whether the death pathways are also the ones dominating at higher above MIC azole concentrations which are responsible for the "effective" antifungal activity. Are there possibly other antifungal effects that could dominate at the concentrations above the MIC?

This concern could be addressed by clarifying the following questions:

- what are the MICs of the respective drugs for these pathogens? I think some data should be presented where the MICs of at least the two azoles used in this study have been tested for *Z. tritici* and *M. oryzae*.

- if the experiments were essentially performed under "sub-MIC" conditions, how does this impact on the conclusion/model?

- can suppressions of apoptosis and/or deletion of *atg4* increase the azole resistance of these pathogens?

If the conclusion is that other apoptosis- and autophagy-independent mechanisms could come into play at concentrations above the MIC, the following statement should be revised:

Discussion, line 508-510: "This finding argues that ergosterol depletion, rather than the proposed accumulation of a toxic sterol intermediate⁸, is responsible for the observed impact in the pathogen cell".

2. It was shown that azoles cause 100% mortality in *Z. tritici* (Fig. 1c). How about *M. oryzae*? Are azoles causing 100% mortality here, too? If not – what impact does it have on the conclusions of the manuscript?

3. Discussion: the results should be discussed in detail in consideration of related work in the literature with similar findings, for example, see above.

Minor points:

Delta-*atg4* mutant: please show growth tests which may reveal any growth defects of this mutant. How was the mutant verified, how many clones were generated and, if any apparent growth defects were found, did all clones show the same behavior?

All figures/investigations - for example, Fig. 3c TMRM fluorescence: How was fluorescence intensity normalized when analyzing and comparing imaging data of different samples/conditions? Was there a treatment-independent fluorescent probe in each sample that has been used to normalize the results?

The manuscript text needs to be thoroughly proofread/revised. Some statements require refinement, and some terms might be only clear to experts and need further explanation (see below for examples).

Is the methodology sound? Does the work meet the expected standards in your field?

Yes, experiments are described in good detail.

Is there enough detail provided in the methods for the work to be reproduced?

Yes.

Needs improvement (examples):

Abstract:

Needs improvement. In its present form it is confusing, containing incorrect statements and incomplete. For example: a) it is not clear which "both programmed cell death pathways" are meant, b) azoles are not ergosterol inhibitors but ergosterol biosynthesis inhibitors, c) "is" missing after action in line 27, d) The sentence "Thus, ergosterol synthesis-targeting fungicides kill crop pathogens by activation of two cell death pathways" does not connect well with the text before.

Manuscript:

Line 37: I suggest defining the term fungitoxic in the context of fungistatic and fungicidal.

Line 46: With respects to 14 α -lanosterol demethylase-linked resistance mechanisms:

overexpression is another common mechanism.

Line 55: the term PM is not explained.

Line 65: the term MoA is not explained.

Line 65: Azoles are not fungicidal against all species: rather antifungal activity instead of fungicidal activity.

Line 81: fungicidal or fungitoxic? Please define fungitoxic (see above)

Line 82: Used

Line 84: Please explain mCherry-Sso1 and His1-ZtGFP. Please explain the roles of the Sso1 and His1 proteins.

Line 88: Why was a concentration of 1 µg per ml selected here?

Line 91: "during 6-8h" what do you mean?

Line 93/94: similar "initially fungistatic, later fungicidal" susceptibility profiles were also found for other fungal pathogens, e.g., *A. fumigatus*.

Line 99: The LIVE/DEAD stain: How is "dead" defined here? Cell membrane integrity failure?

Please explain (and discuss) the mechanisms this staining relies on with respects to what manifestation of death this indicates.

Line 118: what does "EC" stand for?

Line 123: of the outer membrane

Line 149/Fig. 2e: please indicate the Woronin bodies in the electron microscopy images

Line 174: 40% instead of 40.40% and 33% instead of 33.45%?

Line 175: "The gene encoding..." This sentence is incomplete.

Line 186/7: this sentence is incomplete.

Headings of results:

"Azoles alter" and "azoles hyperpolarize" and "azoles induce": this could be misunderstood in a way that azoles cause these effects directly.

Discussion: "This finding argues that ergosterol depletion, rather than the proposed accumulation of a toxic sterol intermediate⁸, is responsible for the observed impact in the pathogen cell. Thus, our study provides novel insight into the MoA of azoles and other inhibitors of the ergosterol biosynthesis pathway. In addition, we show for the first time"

Acquisition of (random) mutations that facilitate resistance requires significant replication taking place. Would the "inhibited" organisms replicate enough so that the mutations occur?

Figures:

- some abbreviations are not explained in the figure legends (e.g., "n.s.")

- Sever figures have "inserts" which were obtained with different microscopic techniques and with different samples (for example, Fig. 4a and Fig. 4c). For me it is often not clear what these are supposed to show. This needs further explanations in the manuscript text and figure legends.

- Fig 7 A: spelling of allylamines

Reviewer #4 (Remarks to the Author):

In this manuscript, the authors investigated the effects of triazoles, epoxiconazole and metconazole, in important fungal pathogens of plants, which significantly affect crop yield. Similar effects were observed in two of these pathogens, *Zymoseptoria tritici* and *Magnaporthe oryzae*. Using a combination of drug experiments and fungal mutant strains, they showed that the azoles are both fungistatic and fungicidal and that these substances induce two pathways of programmed cell death, autophagy and reactive oxygen species-dependent apoptosis. The simultaneous inhibition of both death pathways prevented the azole-induced fungal cell death. This mode of action appears to be shared by other types of ergosterol inhibitors, like the azoles, and this information may be useful for future strategies for fighting fungal pathogens of plants. The authors also found that azoles reduce the fluidity of the plasma membrane, ultimately resulting in the formation of incomplete fungal septa. A few spelling mistakes should be checked throughout. For example, the word "is" is missing in line 27.

Point-by-point response to reviewer comments on NCOMMS-23-45492

Reviewer #1

A) MAJOR POINTS

Is epoxiconazole or metconazole itself degraded into other small molecule metabolites within the fungal cells? The reason for cell death maybe caused by these small molecule metabolites, not the original drug itself. It's important to note that I'm referring to whether the drug itself is metabolized, and not whether the drug causes changes in the fungus' metabolites (such as the mentioned methylsterols in the article). This question is critical because the author does not know whether it is the two compounds themselves or their metabolites that lead to the cell death phenotype.

Our response:

We recently published an extensive transcriptome analysis in *Z. tritici* conidia, treated with epoxiconazole (Cannon et al. 2022, *PLoS Pathogens* **18**:e1010860). This study revealed that epoxiconazole treatment significantly induced the expression of 76 genes encoding metabolising enzymes. Many of these are, indeed, predicted to modify the azole molecule (e.g. Cytochrome P450s and Sulfotransferases). Thus, it is possible that chemically-modified azole molecules inhibit ergosterol biosynthesis. However, such possibility is speculative and, to our knowledge, not supported by research. In any case, epoxiconazole and metconazole are used to control *Z. tritici* in the field and our study shows the effect of these azole fungicides in the pathogen cell. Whether breakdown metabolites are inhibiting ergosterol synthesis or whether the original active ingredients act on 14 α -lanosterol demethylase does not affect this main message.

Epoxiconazole and metconazole are fungicides used to control fungi in plants. Therefore, it is important to consider whether long-term use of these chemicals can lead to plant toxicity and death.

Our response:

We agree with the referee that plant toxicity and death is an important aspect of fungicide use. However, both epoxiconazole and metconazole have been extensively used to control Septoria leaf blotch, caused by *Z. tritici*, in agricultural practise for many years (e.g. Jørgensen et al. 2022, *Front Plant Sci.* **13**:1060428 and literature therein). To our knowledge, no phytotoxic activity of either of these azoles was reported.

Additionally, it is worth investigating if the fungi isolated from treated plants display the same phenotypes under laboratory conditions. I suggest conducting experiments where the plants are treated with the chemicals for 1-2 days, then inoculating them with the fungi and observing any related phenotypic changes.

Our response:

This is a very valid point. Our study aims to address the mode of action of epoxiconazole and metconazole during plant protection, yet the referee is correct in highlighting that we have solely focussed on phenotypes in culture conditions. The major fungicidal modes of action that our study reveals are (1) mROS development, (2) apoptotic cell death and (3) induction of macroautophagy.

Our data suggest that these phenotypes are a consequence of reduced ergosterol levels, as other inhibitors of ergosterol biosynthesis have the same impact on the pathogen cell. In addition, we show in this revised manuscript that high concentrations of azoles affect plasma membrane integrity.

For technical reasons, we cannot use the established staining protocols for mROS generation and apoptotic cell death *in planta*. To overcome this technical challenge, we generated reporter strains that express green fluorescent protein under the promoters of genes, known to be transcriptionally up-regulated when mROS is produced. We have previously used similar promoter-GFP reporter constructs to study gene expression in plant pathogens during host infection (*Ustilago maydis*, Bielska et al. 2014, *Nat. Commun.* 5:5097; *Z. tritici*, Kilaru et al. 2022, *Nat. Commun.* 13:5625).

We have chosen 2 genes for each condition that encode proteins, overexpressed at (i) high mROS levels (Hyr1, Huang et al 2011, *Plos Pathogens*, **7**: e1001335; the mitochondrial superoxide dismutase Sod, Gralla & Kosman 1992, *Adv. Genet.* **30**: 251-319), (ii) apoptosis (the tRNA-cleaving RNase Rny1, Thompson & Parker 2009, *J. Cell Biol.* **185**: 43–50; the metacaspase Mca1, Madeo et al. 2002, *Mol Cell* **9**: 911-917) and (iii) autophagy (the serine-threonine kinase Atg1 and the ubiquitin-like protein Atg8, Bernard et al. 2015, *Autophagy* **11**:2114-2122). In addition, we generated reporter develop a reporter strain for ergosterol depletion (Erg5 and Erg6, both up-regulated in *Z. tritici* upon azole treatment (Cannon et al. 2022, *PLoS Pathogens*, **18**:e1010860). We fused 1500 bp of the promoter region to ZtGFP and integrated these constructs into the *sdi1* locus in *Z. tritici* strain IPO323.

We show in this revised manuscript that the promoters of the chosen candidate genes are significantly induced in the presence of epoxiconazole in liquid culture (control experiments). We also found that all reporter constructs were induced at 2 days after inoculation of wheat plants, pre-treated with epoxiconazole. These results provide strong indication that azoles affect pathogen cells on wheat leaves in a very similar way, they induce mROS formation, apoptosis and autophagy.

In addition, we performed quantitative DiBAC₄(3) and tested if epoxiconazole when sprayed onto wheat leaves depolarises pathogen cells that rest on the plant surface. We found no depolarisation of *Z. tritici* cells resting on epoxiconazole treated wheat leaves.

The description of the experimental procedures, reporter genes, plasmids and strains is included in a Supplementary Table 1, 2, 3, 5, a new Supplementary Table 4, Supplementary Information on plasmid generation (page 20-22), the Methods section (page 28-29, 31-32, 35, 36, 40). All results are included in a new Figure 6c-g and associated figure legend, in the Results (page 16, 17, 18) and in the Discussion (page 26).

The author did not observe the growth status of the mycelium (infectious hyphae) when the pathogen interacts with the host. I suggest inoculating the mycelium after applying the chemicals to the plant for 1-2 days and observing the growth status of the infectious hyphae.

Our response:

This is another valid point. In our study, we focus on the conidia, as these are an infectious inoculum in the field. To meet the referee's request, we have now included data on epoxiconazole-induced mortality in hyphae *in planta*. We show in this revised manuscript that pre-treatment of wheat leaves with 0.1 µg ml⁻¹ epoxiconazole kills *Z. tritici* conidia within 2 days. While hyphae are less affected at this early time point, this azole treatment also kills all hyphal cells within 5 days.

The description of the experimental procedures and the new data is now included in a Supplementary Table 1, 2, 3, the Methods section (pages 35-39). All results are included in a new Figure 6a, 6b and associated figure legend, in the Results (page 17).

B) MINOR POINTS

How is cell death determined in Figure 1a?

Our response:

We have determined cell death in Figure 1c. We apologise for not having been clear in this technical point. We use eGFP-Sso1-expressing conidia, which allowed the identification of individual cells due to the fluorescent labelling of the plasma membrane. We then performed LIVE/DEAD staining and determined the proportion of red-fluorescent (=dead) cells. This is given now as Mortality (= % dead cells). We have included this technical information in Method section of this revised manuscript (page 37/38). We also added a note to the Results (page 4) referring the reader to this information in the Method section.

The control in Figure 1d should also be stained with LIVE/DEAD.

Our response:

The control cells shown in Fig. 1d were stained with LIVE/DEAD™ dye. The images in the original submission showed treatment at 1 µg ml⁻¹. However, in response to reviewer comments, we replaced these data in Figure 1d by LIVE/DEAD™ staining at 4 days at 0.01 µg ml⁻¹ azole treatment (the old data are now shown in Supplementary Fig. 2). In the new image for the control experiment, we now show a single dead cell, stained with the red-fluorescent dye. Thus, although this is a rare event, we show now in the new Fig. 1d that control cells were also stained by the dye.

Is the drug Epoxiconazole or Metconazole used as a raw material or as a formulated product? The text needs to indicate whether it is the active ingredient used.

Our response:

We have added a note to the text (page 4) to make clear that active ingredient was used. This complements the information in the Method part; page 35/36).

Rice blast fungus infects host plants mainly relies on conidia germination and appressorium formation. Compared to the phenotype of vegetable growth, the author should pay more attention to the phenotypes of these two stages.

Our response:

We agree with the reviewer and have now included data on the effect of epoxiconazole and metconazole of germination of conidia and appressorium formation in *M. oryzae*, using an established assay, previously used by us to investigate the effect of fungicides on these developmental processes (Steinberg et al. 2020, *Nat. Commun.* **11**:1608). We found that both azoles

inhibit germination and appressorium formation at sub-lethal concentrations. We also found that metconazole was more effective in inhibition germination of *M. oryzae* than epoxiconazole, yet the reason for this difference is not clear. We included these data in Figure 7m and 7n and accessory legends, in the Results section (page 20), added a new reference (#47) and added a description of the methodology in the Methods section (page 44).

Reviewer #2

Comments:

1) Line87-88: Why did the author use 1 ug/ml azole, not other concentrations? If you use a lower concentration (e.g. 0.01, that was used in Figs. 2-5), what will happen? Is the fungistatic or fungicidal activity related to azole concentration?

Our response:

This is an interesting point that we have now addressed in our revised manuscript. As the use of this concentration was also questioned by other referees (see comment by referee 3), we decided to alter the manuscript here. We now start with determining the minimal inhibitory concentration (MIC) for colony formation on agar plates. We find that the MIC is $\sim 0.01 \mu\text{g ml}^{-1}$ (data included in new Supplementary Fig. 1a, 1b). We then test this concentration for fungistatic and fungicidal activity *in vitro* at this concentration (new Fig. 1b, 1c, 1d). The previous data showing the effect on $1 \mu\text{g ml}^{-1}$ are now shown in a new Supplementary Fig. 2a-c.

We now show that the switch from fungistatic (growth inhibition) to fungicidal (killing of the fungus) is not depending on concentration. However, we find higher mortality in the presence of $1 \mu\text{g ml}^{-1}$ azoles (new Supplementary Fig. 2b). This corresponds with an additional effect of azoles at the 100-times higher concentration on plasma membrane integrity (new Supplementary Fig. 7d), which was found after addressing the request by referee 3 for testing for additional MoAs, at higher concentrations (see below).

In summary, in this revised manuscript, almost all *Z. tritici*-related experiments were done $0.01 \mu\text{g ml}^{-1}$ (which resembles 10% killing after 24h and the MIC on plates), but we also test 100-times higher concentrations of MIC as these might reflect more the activity of azoles when used in the field. These data are now included as a new Supplementary Figs. 1a, 1b, new Fig. 1b-d, Supplementary Fig. 7d), and in the Results (page 4, 5, 17).

2) Fig. 1: Since *Z. tritici* is a filamentous fungus, how to account cell number in this experiment?

Our response:

We apologise that the applied method was not explained sufficiently clear. We use eGFP-Sso1-expressing conidia, which allowed the identification of individual cells due to the fluorescent labelling of the plasma membrane. We then performed LIVE/DEAD™ staining and determined the proportion of red-fluorescent (=dead) cells. This is given now as Mortality (= % dead cells). We have included this technical information in Method section of this revised manuscript (page 37/38). We also added a note to the Results (page 4) referring the reader to this information in the Method section.

3) Line 260-261, and Fig. 5a: What are the eGFP-Atg8-positive autophagocytic organelles? To verify the conclusion that azoles induce autophagy, the authors have to conduct a GFP-Atg8 cleavage analyses with Northern blotting which is a classic method for autophagy research. Importantly, the autophagic process is dynamic, only microscopic observation can not display the global autophagic status in each sample.

Our response:

GFP-Atg8 is considered "a useful marker for macroautophagy" that "remains associated with the complete autophagosome" and thus is "a key autophagy-related marker protein" (Klionsky 2011, *Autophagy* **7**:1093). In contrast to Atg8-GFP, which gets cleaved between Atg8 and the fluorescent tag, GFP-Atg8 is only modified at the C-terminal end which results in conjugation with the phospholipid phosphatidylethanolamine, which is located in phagophores and mature autophagosomes (Martens et al. 2016; *J. Mol. Biol.* **428**:4819). Thus, GFP-Atg8, which we used in this study, is a reliable fluorescent marker for early and late autophagosomal organelles in living cells (Klionsky 2011, *Autophagy* **7**:1093).

Due to its reliable localisation on autophagocytic organelles, this marker was used to study autophagy in a broad range of different organisms. Numerous report the use of GFP-Atg8 homologues, including in protozoa (Mizushima et al. 2014, *Autophagy*, **10**:1487), fungi (e.g. Xie et al. 2008, *Mol. Biol. Cell* **19**:3290; Kilaru et al. 2017, *Fungal Genet Biol.* **105**:16; Kershaw and Talbot, 2009, *PNAS* **106**:15967), insects (Yano & Kurata 2008, *Autophagy* **4**:958; Gai et al. 2013, *Arch. Insect Biochem. Physiol.* **84**:57), mammals and humans (e.g. Meng et al. 2018, *Biomed Pharmacother.* **108**:1617; Zhang et al. 2022, *Methods Mol. Biol.* **24**:2474) and plants (Liu et al. *Front Plant Sci.* **13**:866367). Considering the overwhelming body of literature that uses this GFP-Atg8 to monitor autophagy in living cells, we feel confident that our approach is scientifically sound. We do not see the need or benefit of a "a GFP-Atg8 cleavage analyses with Northern blotting", as suggested by the reviewer.

However, we provide now more direct evidence that GFP-Atg8 positive organelles are engulfing nuclei (labelled with red-fluorescent histone 1). This result was already implied by electron microscopy images in the previous version of the manuscript and is now support by these live cell imaging result. These new data fully support the notion that the GFP-Atg8 structures are autophagosomes. All results are now included in the Results section (page 12), Figure 5f, Supplementary Fig. 6b and additional information on the newly generated strain and plasmid is included in Supplementary Table 1-3, Supplementary Information on Plasmid Generation, Methods (page 31). In addition, 2 new references were included (#40, #41). The previous electron microscopy image in Fig. 6 moved to Supplementary Fig. 6c.

4) Lines 295-298: The authors found that $\Delta atg4$ mutants were strongly reduced in azole-induced mortality. Would induction of autophagy (such as nitrogen starvation treatment) increase azole-induced mortality?

Our response:

This is an interesting question raised by the reviewer. To address this point, we firstly confirmed that nitrogen starvation increases the number of autophagosomes in *Z. tritici*. We next starved cells and subsequently treated them with epoxiconazole for 24h. This was followed by quantitative analysis of

LIVE/DEAD™ staining. We found that starvation increased mortality in epoxiconazole-treated cells and cell exposed to the solvent only. However, even when epoxiconazole-induced mortality was corrected for the "background mortality", found in starving solvent-treated cell, nitrogen starvation was still significantly increased the azole-induced mortality. These data are now included in Supplementary Fig. 6g, 6h and the Results section (page 14). The methodology was included in the Methods (page 27/28).

5) Line 312-313: *The $\Delta atg4$ mutants still formed a low number of phagophores, and 25 μM 3-MA was unable to block autophagy completely, the author should test sensitivity of $\Delta atg8$ to azole since *Agt8* is essential for autophagy in fungi.*

Our response:

We thank the reviewer for this suggestion. We agree that autophagy is not completely blocked and small GFP-Atg8 structures are still visible in $\Delta atg4$ mutants and 3-MA-treated cells. However, epoxiconazole did not induce autophagosome formation in $\Delta atg4$ mutants (Fig 5h; 5i) and 3-MA prevented the formation of large autophagic structures (Fig. 5k). Moreover, 3-MA brought down epoxiconazole-induced mortality by ~35% (Fig. 5l).

It is likely that these effects are increased by establishing $\Delta atg8$ mutants in *Z. tritici*. However, this would also require establishing another fluorescent marker for autophagic organelles, as we currently use GFP-Atg8 to visualise these structures (see also point about *atg4* mutant below). Moreover, we would have to establish a transformation system for *M. oryzae* (the current $\Delta atg4$ strain was a gift from Prof. Nick Talbot). We consider these technical challenges manageable but feel that the benefit from these advances do not fully justify the experimental effort. This conclusion was agreed with the editor of this submission.

6) Fig 8. *Is *Z. tritici* able to take up exogenous ergosterol? (*S. cerevisiae* cells can take up exogenous sterols under anaerobic conditions). The logic of Fig. 8 shows that septation and cell death are depended on ergosterol biosynthesis. Is this true absolutely? If *Z. tritici* can take up exogenous ergosterol, when *Z. tritici* is supplemented with exogenous ergosterol, can azoles still induce cell death and incomplete septation in the fungi?*

Our response:

We appreciate this point by the referee. We performed the requested experiments by applying exogenous ergosterol to epoxiconazole-treated cells of *Z. tritici*. We found that the application of exogenous ergosterol did significantly reduce the (a) septation defect, (b) the formation of autophagosomes and (c) the mitochondrial potential (c). However, compared to control cells (no epoxiconazole treatment) mitochondria were still hyperactive and thus mROS was still formed. This, in turn, induced apoptosis. Thus, our results show partial rescue of azole-induced defects. However, none of the phenotypes was completely suppressed by exogenous ergosterol. We consider it is possible that the partial rescue of epoxiconazole-induced defects correlates with low uptake of exogenous ergosterol. Moreover, this acquired exogenous ergosterol may be trapped in the plasma membrane and other organelle membranes, which further hinders accumulation in the inner mitochondrial membrane, where the apoptosis-inducing mROS is formed. We include all data in the Results (page 21/22) and a new Supplementary Figure 11a-e; the methodology is described in the Methods part (page 36).

Reviewer #3

Major points:

1. Many experiments were performed at very low azole concentrations (0.01 $\mu\text{g per ml}$) which are presumably (?) below the minimal inhibitory concentrations (MICs) of the drugs. (Line 112-114: "Previous work has shown that the primary physiological effect of fungicides can be determined at low concentrations, where mortality is at ~10-20% and the majority of the cells are still alive").

Our response:

This is a good point. The minimal inhibitory concentration (MIC) is defined as "the lowest concentration of an antimicrobial that will inhibit the visible growth" (Andrews 2001, *J. Antimicrob. Chemother* **48**:5-16). We show in Fig. 1 that azoles inhibit growth without killing the fungal cell. Thus, the MIC is not necessary directly related to the concentration at which we find 10% pathogen mortality (0.01 μgml^{-1}). However, we agree with the referee that point needs attention to avoid confusions.

In this revised version of our manuscript, we have now determined the MIC of epoxiconazole and metconazole in *Z. tritici*. It turns out that both MICs are almost 0.01 $\mu\text{g ml}^{-1}$ in *Z. tritici* (metconazole: $\sim 0.007 \mu\text{g ml}^{-1}$; epoxiconazole: $\sim 0.011 \mu\text{g ml}^{-1}$). The data are shown in a new Supplementary Fig. 1a, 1b and are included in the Results section (page 4, 6, 16).

While the experimental results obtained under these conditions (apoptosis/autophagy) appear sound and are interesting, it raises the question whether the death pathways are also the ones dominating at higher above MIC azole concentrations which are responsible for the "effective" antifungal activity. Are there possibly other antifungal effects that could dominate at the concentrations above the MIC?

Our response:

This is an interesting point. Indeed, disruption of the plasma membrane was suggested to be the main MoA by others. To address this comment, we have now performed all essential MoA studies at 1 $\mu\text{g ml}^{-1}$, a concentration that is ~ 100 -times higher than the MIC (see comment above). We show that both azoles still induce mROS, apoptosis and autophagy at this high concentration (new Supplementary Fig. 7a-c). Interestingly, at this high concentration, the integrity of the plasma membrane is also affected (new Supplementary Fig. 7d), an effect that was not found at 0.01 $\mu\text{g ml}^{-1}$ (Fig. 1i). This additional fungicidal impact explains the increased mortality of *Z. tritici* cells at 1 $\mu\text{g ml}^{-1}$ ($\sim 100\%$ after 4 days; data now shown in new Supplementary Fig. 2b, 2c), compared to treatment with 0.01 $\mu\text{g ml}^{-1}$ (now shown in new Fig. 1 c, 1d).

In response to referee 1, we tested the MoAs of epoxiconazole on the plant surface (we found that mROS, apoptosis and macroautophagy is induced; see above). Realising that 1 $\mu\text{g ml}^{-1}$ compromised plasma membrane integrity *in vitro*, we tested for such MoA on wheat leaves that were pre-treated with 0.1 and 1 $\mu\text{g ml}^{-1}$ epoxiconazole. These experiments provided no indication that cells are depolarised when wheat plants were pre-treated with azoles. Thus, we consider it most likely that the fungicidal activity of azoles at higher concentrations is due to the same MoA.

In summary, we now tested all fungicidal MoAs at 0.01 and 1 $\mu\text{g ml}^{-1}$, this revealed that impaired plasma membrane integrity is a new MoA at higher azole concentrations, which is restricted to cells

grown in liquid culture. These results are now included in a new Fig. 6g and associated legend, Supplementary Fig. 7., in the Results section (page 5, 16/17,) and are discussed in the Discussion section (page 22, 26) .

This concern could be addressed by clarifying the following questions:

*- what are the MICs of the respective drugs for these pathogens? I think some data should be presented where the MICs of at least the two azoles used in this study have been tested for *Z. tritici* and *M. oryzae*.*

Our response:

We have performed the requested experiments and now included data on the MIC of epoxiconazole and metconazole for *M. oryzae* and *Z. tritici* in the revised version of this manuscript. We essentially show that the *Z. tritici* experiments were done at "MIC conditions" (*Z. tritici* MICs are 0.007-0.011 $\mu\text{g ml}^{-1}$, experiments were done at 0.01 $\mu\text{g ml}^{-1}$; *M.oryzae* MICs are 2 - 6 $\mu\text{g ml}^{-1}$, experiments were done at 10 $\mu\text{g ml}^{-1}$). The data are shown now in a new Supplementary Fig. 1a-d and in the Results section (page 4, 18/19)

- if the experiments were essentially performed under "sub-MIC" conditions, how does this impact on the conclusion/model?

Our response:

As outlined above, all experiments were either performed at "MIC conditions" or "above-MIC conditions". Thus, the much-appreciated concern of the referee that experiments were done under "sub-MIC conditions" is no longer valid.

*- can suppressions of apoptosis and/or deletion of *atg4* increase the azole resistance of these pathogens?*

Our response:

These are important questions. Indeed, our manuscript already shows that $\Delta atg4$ mutants are significantly less sensitive to epoxiconazole (now Fig. 5j, Fig. 7l); to address the sole role of apoptosis, we performed time course experiments in which we co-treat *Z. tritici* cells with epoxiconazole and rotenone, followed by daily analysis of cell mortality (Fig. 5l; green curve). As rotenone prevents mROS development and apoptosis (Fig. 4i, 4j), this treatment allows an estimation of the contribution of mROS/apoptosis and macroautophagy to the fungicidal activity of epoxiconazole. We found that ~30% of all cells still died under these conditions, confirming that autophagy significantly contributes to the fungicidal activity of epoxiconazole.

To clarify this point further, we provide now a new graph (Fig. 5n), in which the mortality of epoxiconazole after 5 days was corrected for the "background mortality" in control experiment. The

remaining mortality was set to 100%, illustrating the maximum killing potential of epoxiconazole. When this was then compared to "background mortality"-corrected data for epoxiconazole+rotenone, epoxiconazole+3-MA and epoxiconazole+rotenone+3-MA-treated *Z. tritici* cells, it became very clear that mROS- and induced apoptosis contributes ~54% of the fungicidal activity of epoxiconazole (Fig. 5n). On the other hand, blocking mROS formation (and thus apoptosis) left ~30 % of the potential of epoxiconazole. When both autophagy and apoptosis is impaired, mortality in the presence of epoxiconazole drops to ~4%. This result clearly demonstrates that both, apoptosis and autophagy contribute to the fungicidal activity of epoxiconazole.

These new results are included in the revised manuscript (Fig. 5l and Fig. 5n; Results section, page 15, 16; Discussion page 25/26).

If the conclusion is that other apoptosis- and autophagy-independent mechanisms could come into play at concentrations above the MIC, the following statement should be revised:

Discussion, line 508-510: "This finding argues that ergosterol depletion, rather than the proposed accumulation of a toxic sterol intermediates, is responsible for the observed impact in the pathogen cell".

As outlined above, we did find a new azole MoA at higher concentration (depolarisation of the plasma membrane as an indication of an effect on the integrity of the plasma membrane). However, this MoA appears to be restricted to *in vitro* applications in liquid culture.

We think that the statement, cited by the referee, is still correct. This conclusion is based on the result that other ergosterol inhibitors cause the same defects in septation, mitochondrial hyperactivation, mROS development and induction of apoptosis and autophagy as azoles do. If these cellular responses would be due to toxic intermediates that are formed due to the specific inhibition of 14 α -lanosterol demethylase (as proposed by Kelly et al. 1995, *Biochem. Biophys. Res. Commun.* **207**:910-915.), inhibition at other enzymatic steps in the biosynthetic ergosterol pathway should not cause the same phenotypes. The logic behind this is that inhibition of other biosynthetic enzymes (e.g. squalene epoxidase by terbinafine) would result in different intermediates, which are unlikely to have the same toxic activity in the pathogen cell. This logical conclusion is independent of any potential "above MIC" impact of azoles (for which we have no supportive evidence anyway).

Following these arguments, we decided to keep the statement. However, we explained our argument better now and also now briefly discuss the possibility that so far undetected MoAs at high concentrations of azoles could contribute to their anti-fungal activity *in planta* (Discussion section, page 26).

2. It was shown that azoles cause 100% mortality in Z. tritici (Fig. 1c). How about M. oryzae? Are azoles causing 100% mortality here, too? If not – what impact does it have on the conclusions of the manuscript?

Our response:

We agree with the referee and have included data on *M. oryzae* mortality in the presence of epoxiconazole and metconazole in the revised manuscript. From these data we conclude that azoles are fungicidal in *M. oryzae*, yet they are much less potent in this pathogen. We included these results in a new Supplementary Fig. 8 and in the Result section (page 19).

3. Discussion: the results should be discussed in detail in consideration of related work in the literature with similar findings, for example, see above.

Our response:

We agree with the referee that careful citation is vital. We thought we did this already in the original manuscript (and so thought another reviewer). Indeed, one suggested reference was already included in the first version of the manuscript (Lee and Lee 2018). The references highlighted as missing are all dealing with mROS development in several human pathogens. Due to space restrictions by the journal, these references were initially not cited in the context of mROS development, as we focussed in the discussion of *C. albicans* and *S. cerevisiae*, for which he had included citations (Kobayashi et al. 2002, Belenky et al. 2013). However, we appreciate the referee's request and now included Shekhova et al., 2017 and Ferreira et al. 2013, showing results in two other human pathogens. Having covered the fact that mROS development and apoptosis was previously found in fungi by citing 4 references appears enough to make the point that we were not the first to show that azoles induce mROS in fungi. We include the new references (#18; #64) and highlight these references in the Introduction (page 3) and the Discussion (page 24).

Minor points:

Delta-atg4 mutant: please show growth tests which may reveal any growth defects of this mutant.

Our response:

We tested for growth on agar plates and for morphology defects in liquid culture. We found no defects in 3 independent *atg4* deletion mutant strains. We have included these results in this revised manuscript (new Supplementary Fig. 6e, 6f; Results section, page 13).

How was the mutant verified, how many clones were generated and, if any apparent growth defects were found, did all clones show the same behavior?

Our response:

The mutant was verified by Southern blot and explained in the original manuscript (page 26; now page 31/32). Three clones of $\Delta atg4$ mutants were generated and tested for morphology and growth defects (see text above). As said above, no growth or morphology defects were found. We have included these results in this revised manuscript (new Supplementary Fig. 6e, 6f; Results section, page 13).

All figures/investigations - for example, Fig. 3c TMRM fluorescence: How was fluorescence intensity normalized when analyzing and comparing imaging data of different samples/conditions?

Our response:

When images were compared (e.g. Fig. 3c), microscopic settings were kept identical (e.g. laser output, exposure time, binning, gain). Images were processed identically (same images scaling, gamma, contrast and brightness). We have now added a note to all figure legends to emphasise this (Fig. 3c, 4a, 6c, 6e).

Was there a treatment-independent fluorescent probe in each sample that has been used to normalize the results?

Our response:

For quantitative comparisons, each replicate set of experiments was performed with its own solvent control, containing the identical amount of methanol or DMSO, but lacking azoles or inhibitors. TMRM and DHR123 intensities were measured in unprocessed raw 14-bit images, taken under identical microscopical settings, and absolute values were plotted (e.g. Fig. 3d).

The newly added MoA reporter-expressing strains, used to analyse the response of *Z. tritici* to epoxiconazole on wheat leaves and in liquid medium (see new Figure 6) were exposed to the solvent methanol (=control) and to the fungicide. The cytoplasmic background fluorescence in cells, grown in control experiments, was set to 100% and the fluorescent signal in cells exposed to the azoles was related to this value. This information was added to the legends of Fig. 6d and 6f.

The manuscript text needs to be thoroughly proofread/revise. Some statements require refinement, and some terms might be only clear to experts and need further explanation (see below for examples).

Our response:

We thank the reviewer for this helpful comment. We have addressed and correct all concerns, listed below, and add additional explanations where requested in the revised version of the manuscript (see below).

Abstract:

Needs improvement. In its present form it is confusing, containing incorrect statements and incomplete. For example: a) it is not clear which "both programmed cell death pathways" are meant, b) azoles are not ergosterol inhibitors but ergosterol biosynthesis inhibitors, c) "is" missing after action in line 27, d) The sentence "Thus, ergosterol synthesis-targeting fungicides kill crop pathogens by activation of two cell death pathways" does not connect well with the text before.

We have revised the abstract and addressed all suggestions.

Manuscript:

Line 37: I suggest defining the term fungitoxic in the context of fungistatic and fungicidal.

The term "fungitoxic" is not a common term and we therefore replaced it by "anti-fungal". We also defined fungistatic and fungicidal more clearly (page 2/3) and added a new reference to support these definitions (#9, Graybill et al. 1997).

Line 46: With respects to 14 α -lanosterol demethylase-linked resistance mechanisms: overexpression is another common mechanism.

We agree here entirely. While our initial statement is correct ("mutations in the enzyme cause resistance"), such statement, indeed, initiates a discussion about resistance mechanisms. This aspect is not of relevance for this manuscript. We therefore have replaced our initial statement regarding "resistance in this enzyme" by "which is a key enzyme in the ergosterol biosynthetic pathway" (see Introduction, page 2)

Line 65: the term MoA is not explained.

We replaced MoA here by "effect" and define MoA at the end of the Introduction (page 3).

Line 65: Azoles are not fungicidal against all species: rather antifungal activity instead of fungicidal activity.

Text was changed as suggested (page 3).

Line 81: fungicidal or fungitoxic? Please define fungitoxic (see above)

This good point by the referee was already addressed. We defined fungistatic and fungicidal more clearly (page 2/3) and added a new reference to support these definitions (#9, Graybill et al. 1997). To avoid confusion, the term fungitoxic was removed from the manuscript.

Line 82: Used

Text was changed as suggested (page 3)

Line 84: Please explain mCherry-Sso1 and His1-ZtGFP. Please explain the roles of the Sso1 and His1 proteins.

We changed the text as requested and provide more explanation (Results, page 4).

Line 88: Why was a concentration of 1 $\mu\text{g per ml}$ selected here?

We admit that we selected this concentration randomly. However, following this reviewer's request (see above) and comments made by reviewer 2, we altered the manuscript here. We now start with determining the minimal inhibitory concentration, which is 0.01 $\mu\text{g ml}^{-1}$ (data included in new Supplementary Fig. 1a, 1b). We then test this concentration for fungistatic and fungicidal activity *in vitro* (new Fig. 1b, 1c, 1d) and move the data for test 1 $\mu\text{g ml}^{-1}$ to a new Supplementary Fig. 2. This reveals that the switch from fungistatic (growth inhibition) to fungicidal (killing of the fungus) is not depending on concentration. However, we find higher mortality in the presence of 1 $\mu\text{g ml}^{-1}$ azoles (new Supplementary Fig. 2b). This corresponds with an additional effect of azoles at the 100-times higher concentration of MIC on plasma membrane integrity (new Supplementary Fig. 7d), which was found after addressing the request by referee 2 for testing the MoA at higher concentrations (see above).

In summary, we now focus on 0.01 $\mu\text{g ml}^{-1}$ as this concentration reflects the MIC (Supplementary Fig. 1a, 1b). We test, however, if 1 $\mu\text{g ml}^{-1}$ is also fungistatic as this high concentration is likely to be more representative of the amount of azoles used in the field (Hof [2001] reports 10 mg per square meter field; reference now included in the text). Indeed, we find an increased fungicidal activity *in vitro* at this high concentration. These data for are now in a new Supplementary Figure 2. We also find at this high concentration that azoles affect plasma membrane integrity (New Supplementary Fig. 7d). Thus, the increased mortality of azoles at 1 $\mu\text{g ml}^{-1}$ correlates with the appearance of an additional MoA (inducing a depolarisation of the plasma membrane) at this concentration. This new MoA was found after we addressed a later point made by this referee point, which essentially asked if higher concentration reveal the same and/or other MoAs (see below). This effect was not seen in pathogen cells that were resting on epoxiconazole-treated wheat leaves. Thus, we consider it likely that this MoA of azole is restricted to treatment in liquid culture and is not of major relevance for the fungicidal activity of azoles in plants. We included this data in the new Figure 6g (Results, page 14; Discussion, page 22).

All data summarised here are now included in a new Supplementary Figures 1a, 1b, main new Figure 1b, 1c, 1d, new Supplementary Fig. 2a-c (former Fig. 1b-d), new Supplementary Figure 7d. A new reference was added (#9) and the text was changed in the Results section (page 4, 5, and 18) and the Discussion on page 26.

Line 91: "during 6-8h" what do you mean?

We changed the text to correct the mistake (page 5).

Line 93/94: similar "initially fungistatic, later fungicidal" susceptibility profiles where also found for other fungal pathogens, e.g., A. fumigatus.

The fungicidal activity in *A. fumigatus* was already mentioned in the text. We have now highlighted that the same susceptibility profile is found in this fungus (page 5).

Line 99: The LIVE/DEAD stain: How is "dead" defined here? Cell membrane integrity failure? Please explain (and discuss) the mechanisms this staining relies on with respects to what manifestation of death this indicates.

We have added a note explaining the mechanism by which this dye allows detection of dead cells (page 4/5)

Line 118: what does "EC" stand for?

EC stands for effective concentration. We removed this term from the main text (page 6) . This now explained in the legend for Fig. 1f, Fig. 7a.

Line 123: of the outer membrane

Text was corrected accordingly (page 6).

Line 149/Fig. 2e: please indicate the Woronin bodies in the electron microscopy images

Fig. 2e and its figure legend were altered accordingly.

Line 174: 40% instead of 40.40% and 33% instead of 33.45%?

We went for a compromise and altered the number to 40.4% and 33.5%. We also stood with this format for all other percentage values in this revised manuscript.

Line 175: "The gene encoding..." This sentence is incomplete.

We apologise for this mistake. The text was rewritten, and the error was erased (page 8).

Line 186/7: this sentence is incomplete.

We have rewritten this sentence in this revised version of the manuscript (page 9).

Headings of results:

"Azoles alter" and "azoles hyperpolarize" and "azoles induce": this could be misunderstood in a way that azoles cause these effects directly.

We see the reviewer's point. We changed most headings, now saying "Azole treatment alters..." rather than "Azole alter..."

Discussion: "This finding argues that ergosterol depletion, rather than the proposed accumulation of a toxic sterol intermediate⁸, is responsible for the observed impact in the pathogen cell. Thus, our study provides novel insight into the MoA of azoles and other inhibitors of the ergosterol biosynthesis pathway. In addition, we show for the first time"

We are not clear what the referee is asking/highlighting here.

Acquisition of (random) mutations that facilitate resistance requires significant replication taking place. Would the "inhibited" organisms replicate enough so that the mutations occur?

We are not clear what the referee is asking here. However, we are not discussing resistance development in the current version of the manuscript, thus do believe that the question raised is not relevant for our revised submission.

Figures:

- some abbreviations are not explained in the figure legends (e.g., "n.s.")

We apologise for this shortcoming. We now went through the Figure legends and explained all abbreviations (incl. n.s., asterisks, or protein names, such as Atg8, Sso1 or His1)

- Sever figures have “inserts” which were obtained with different microscopic techniques and with different samples (for example, Fig. 4a and Fig. 4c). For me it is often not clear what these are supposed to show. This needs further explanations in the manuscript text and figure legends.

Only the inset in Fig. 4c and Supplementary Fig. 5b were obtained by a different method. We have now separated this insert from the panel figure and made them sub-panel Fig. 4d and Supplementary Fig. 5c. All following sub-panels were renumbered accordingly. All other insets (Fig. 2a, 4a, 5m; Supp. Fig. 5d, 5e, 6c) were obtained with the same technique as the main image. A better description of these insets is now added to the respective figure legends.

Fig 7 A: spelling of allylamines

The mistake was corrected.

Reviewer #4

A few spelling mistakes should be checked throughout. For exemple, the word “is” is missing in line 27.

Our response:

We have corrected the mistake and proof-read the manuscript to remove spelling mistakes.

Reviewer #1 (Remarks to the Author):

I have no further comments regarding of the revised version.

Reviewer #2 (Remarks to the Author):

I think that the reviewers have addressed most concerns raised by reviewers. I don't have additional comments on this revision.

Reviewer #3 (Remarks to the Author):

The authors have greatly addressed all my questions. Great work.